# Early-stage lung adenocarcinoma affects DNA methylation and gene expression in adjacent tissues

Yifan Wu [1,13], Yadong Wang [2,3,13], Yao Tang [4,5,6,13], Jianchao Xue [2,3,13], Zichen Jiao [1,4], Bowen Li [2,3], Sainan Wang [1], Zhicheng Huang [2,3], Xiaoyi Zheng [1], Chenzheng Guan [1], Daoyun Wang [2,3], Ji Li [7], Lan Song [8], Ka Luk Fung [2,9], Heqing Xu [2,10], Shanqing Li [2], Liucun Zhu [11], Jian-Qun Chen [1], David J Kerr [12], Naixin Liang [2✉], Qiang Wang [1✉] & Qihan Chen [5,6✉]

## Abstracts

The impact of early-stage tumors on gene expression in adjacent tissues remains uncertain, despite the known influence of the tumor microenvironment on tumor progression. Here, we systematically analyze early-stage lung adenocarcinoma (LUAD) and surrounding tissues across multiple distinct regions, from the tumor core to distant tissues. DNA methylation profiling in a 12-patient cohort reveals two distinct patterns of methylation changes. Steep changes occurring at the tumor boundary and shallow changes showing a gradual shift over increasing distance to the tumor. Approximately 17,000 CpG sites demonstrate shallow changing trends without clear boundaries, potentially affecting 2655 genes. In half of the patients, tissues within 10 mm beyond the tumor show methylation patterns similar to tumors. We test mRNA expression of key genes affected by these methylation patterns and observe that the protein expression pattern of WNT7B demonstrates no steep changes at the tumor boundary, supporting their regulatory role. Adding a 59-patient four-year-prognosis cohort allowed us to rigorously assess the clinical relevance of these methylation change trends. These shallow changes reflect tumor characteristics and have the potential for prognostic prediction in patients, warranting further investigation.

**Keywords** Lung Adenocarcinoma; Tumor Boundary; DNA Methylation; mRNA Expression; Prognosis Prediction
**Subject Categories** Biomarkers; Cancer; Chromatin, Transcription & Genomics

## Introduction

The process of carcinogenesis and tumor progression has benefited from comparisons of tumor and normal tissues. Compared with adjacent normal tissues, solid tumor tissues exhibit greater variation in cellular size, a more complex structural composition, and distinctive stromal components (Chen et al, 2015; Dvorak, 1986; Dvorak et al, 1981; Folkman and Shing, 1992; Komura and Ishikawa, 2018). Consequently, tumor tissues display visually distinct boundaries from surrounding normal tissues, as evidenced by conventional hematoxylin–eosin (HE) staining of pathological sections (Acs et al, 2019; Wang et al, 2021a). However, recent research has indicated that tissues outside the visible tumor boundary may play an important role in tumor development, potentially undergoing varying degrees of alteration and contributing to the tumor microenvironment. The tumor microenvironment serves as a crucial interface with cancer cells and is involved in immune evasion, drug resistance, and promotion of metastasis (Barkley et al, 2022; Cazet et al, 2018; Dirkse et al, 2019; Puram et al, 2017). Transcriptome analysis revealed that, compared with healthy tissues, normal tissues adjacent to multiple tumors activate specific genes associated with inflammatory responses (Aran et al, 2017). Dedifferentiated signatures and stem-cell-like features have been observed in parenchymal cells neighboring cancer cells (Ombrato et al, 2019). Therefore, tumors may possess more extensive molecular boundaries than previously understood.

In recent years, omics-based studies aimed at assessing tissue status have gained considerable popularity. Among these, DNA methylation has emerged as a critical target due to its role as a specific marker for identifying various tissues and cells capable of differentiating between tissue identities, as demonstrated by numerous studies (Loyfer et al, 2023; Teschendorff et al, 2020; Zhu et al, 2022). Such methylation information has paved the way for the rapid development of early screening and diagnostic techniques based on cell-free DNA (cfDNA) methylation in tumor

[1]The State Key Laboratory of Pharmaceutical Biotechnology, School of Life Sciences, Nanjing University, Nanjing, China. [2]Department of Thoracic Surgery, Peking Union Medical College Hospital, Chinese Academy of Medical Sciences, Beijing, China. [3]Chinese Academy of Medical Sciences and Peking Union Medical College, Beijing, China. [4]Department of Thoracic Surgery, Nanjing Drum Tower Hospital, The Affiliated Hospital of Nanjing University Medical School, Nanjing, China. [5]Faculty of Health Sciences, University of Macau, Macau SAR, China. [6]MOE Frontier Science Centre for Precision Oncology, University of Macau, Macau SAR, China. [7]Department of Pathology, Peking Union Medical College Hospital, Chinese Academy of Medical Sciences, Beijing, China. [8]Department of Radiology, Peking Union Medical College Hospital, Chinese Academy of Medical Sciences, Beijing, China. [9]Faculty of Arts and Sciences, Pharmaceutical Chemistry Specialist, University of Toronto, Toronto, Canada. [10]Faculty of Social Sciences and Law, University of Bristol, Bristol, UK. [11]School of Life Sciences, Shanghai University, Shanghai, China. [12]Radcliffe Department of Medicine, University of Oxford, Oxford, UK. [13]These authors contributed equally: Yifan Wu, Yadong Wang, Yao Tang, Jianchao Xue. ✉E-mail: liangnaixin@pumch.cn; wangq@nju.edu.cn; lyonchen@umac.mo

research (Liang et al, 2022; Tse et al, 2021; Yu et al, 2021). In addition, DNA methylation plays a crucial role in gene regulation and is involved in cancer development. For example, methylation of the promoters of tumor suppressor genes or oncogenes may be a key driver of tumorigenesis (Dawson and Kouzarides, 2012; Greger et al, 1989; Herman et al, 1995; Kong et al, 2019; Merlo et al, 1995), and some oncogenes are also activated by DNA hypermethylation in specific genomic regions (Su et al, 2018). The perspective of DNA methylation, therefore, offers a clearer understanding of tumor and surrounding tissue state changes, with important implications for our comprehension of cancer.

In 2021, Jia et al made a noteworthy discovery in lung cancer research by identifying substantial changes in methylation at specific sites along the visual boundaries of lung cancer samples Jia et al (2021). They evaluated the impact of tumors on adjacent tissues from a novel perspective, offering a fresh concept for future research. However, owing to the limitations of samples, sites, and study methods, this study may not be able to comprehensively evaluate the relationships between tumor tissues and adjacent tissues. Despite relatively lenient filter standards for methylation changes, this study raises an important question regarding whether these modification-changed sites/regions with gradual changes across visual boundaries are indeed involved in the regulation of relevant genes and influence tumor development.

To further explore this discovery, we plan to employ various regions of surgical samples from early LUAD patients to systematically evaluate DNA methylation in a discovery cohort and the mRNA expression of corresponding genes in a validation cohort. We will select shallow-changing sites to further investigate their potential for LUAD prognosis by analyzing the methylation status of tumor and adjacent tissues. This study aimed to investigate the impact of tumors on adjacent tissues beyond the visual boundary at the molecular level and to identify potential critical targets involved in LUAD development.

# Results

## Study design

To investigate the potential impact of tumor tissue on adjacent tissues, we conducted a three-stage study, as illustrated in Fig. 1A. In the discovery stage, we collected surgical tissues from 12 early-stage LUAD patients in the Department of Thoracic Surgery. The tissues were then divided into seven regions based on visual boundaries: TC (tumor core), TE (tumor edge), P5 (paired-adjacent 0–5 mm), P10 (paired-adjacent 5–10 mm), P15 (paired-adjacent 10–15 mm), P20 (paired-adjacent 15–20 mm) and PN (paired-distal normal) (Fig. 1B), in accordance with the criteria of the previous study (Jia et al, 2021). An example of a tissue sample is shown in Fig. 1C. All the samples were confirmed by hematoxylin–eosin (HE) staining to determine their pathology and the location of tumor boundaries. A set of samples from a patient is shown in Fig. 1D as an example, where the morphology and staining patterns of TC and TE cells differ markedly from those of the P5–PN regions, with a distinct visual boundary observed outside the TE tissue. Subsequently, we extracted DNA from each sample of the 12 patients and obtained the DNA methylation information and status by reduced representation bisulfite sequencing (RRBS).

Based on the analysis, we selected five genes with significant trends in DNA methylation as targets for further expression testing in another group of early-stage LUAD patients, serving as a validation cohort. Similarly, samples from another 24 early-stage LUAD patients from the Department of Thoracic Surgery were collected, divided, and confirmed following the same procedures. We analyzed the expression of the five target genes in each patient sample using quantitative reverse transcription PCR (RT-qPCR). Detailed information about patients is provided in Table EV1.

## DNA methylation of 12 patients in the discovery stage

We performed quality control and alignment of RRBS data obtained from 84 tissue samples from 12 patients. A total of 4,343,110 CpG sites were detected with coverage greater than 15×. We recorded the frequency of coverage for each site across all 84 samples and plotted the resulting frequency distribution (Appendix Fig. S1A). To include as many CpG sites as possible while ensuring their detection in most samples, we selected 1,530,593 CpG sites for subsequent research, all of which were effectively detected in at least 68 tissue samples (Appendix Fig. S1B). Although some patients may have missing methylation information at certain sites in some samples, this does not affect our overall analysis.

Using the available data, we investigated whether an ordered methylation pattern exists at the CpG sites of tissues from different locations. To objectively evaluate this pattern, we employed the following statistical methods. Detailed information on data processing, filtering, and classification can be found in the Methods section and the Appendix Methods. Briefly, for each CpG site, the methylation levels in seven different regions were initially quantified. Subsequently, a comparison was made between each pair of regions to assess whether significant differences existed at that site. The results of these comparisons were used to construct a $7 \times 7$ comparison matrix, where each cell indicated whether a significant difference in methylation status existed between the corresponding two regions. We also employed the spatial autocorrelation measure Moran's $I$ to further evaluate whether the matrix exhibited observable regularity. A higher Moran's $I$ value indicates greater regularity (Fig. EV1A). Here, we present three examples: chr1:15754767 site exhibited significantly lower methylation levels between TC/P10, TC/PN, P5/P10, P5/PN, and P20/PN, while the comparison between P10/P20 showed the opposite trend. The Moran's $I$ value was relatively small and showed no obvious pattern (Fig. 2A). The methylation level of chr12:51820212 was significantly lower in TC compared to P5-PN, TE compared to P20 and PN, P5 compared to PN, and P10 compared to PN, respectively, and Moran's $I$ was 0.1581, indicating a clear pattern of change (Fig. 2B). The site chr6: 36681559 exhibited significantly lower methylation levels in TC and TE compared to P5-PN. The Moran's $I$ value was higher, indicating a more significant difference in tumor-normal tissue changes (Fig. 2C).

We identified a total of 613,724 CpG sites, accounting for approximately 40% of all selected CpG sites, that showed significant differences between at least two region pairs. To assess the regularity of these sites, we employed the aforementioned method to illustrate the differences among them and presented their Moran's $I$ distribution patterns in blue (Fig. 2D). As a control, we simulated Moran's $I$ distribution pattern of sites under conditions of complete random variation and presented it in brown. The

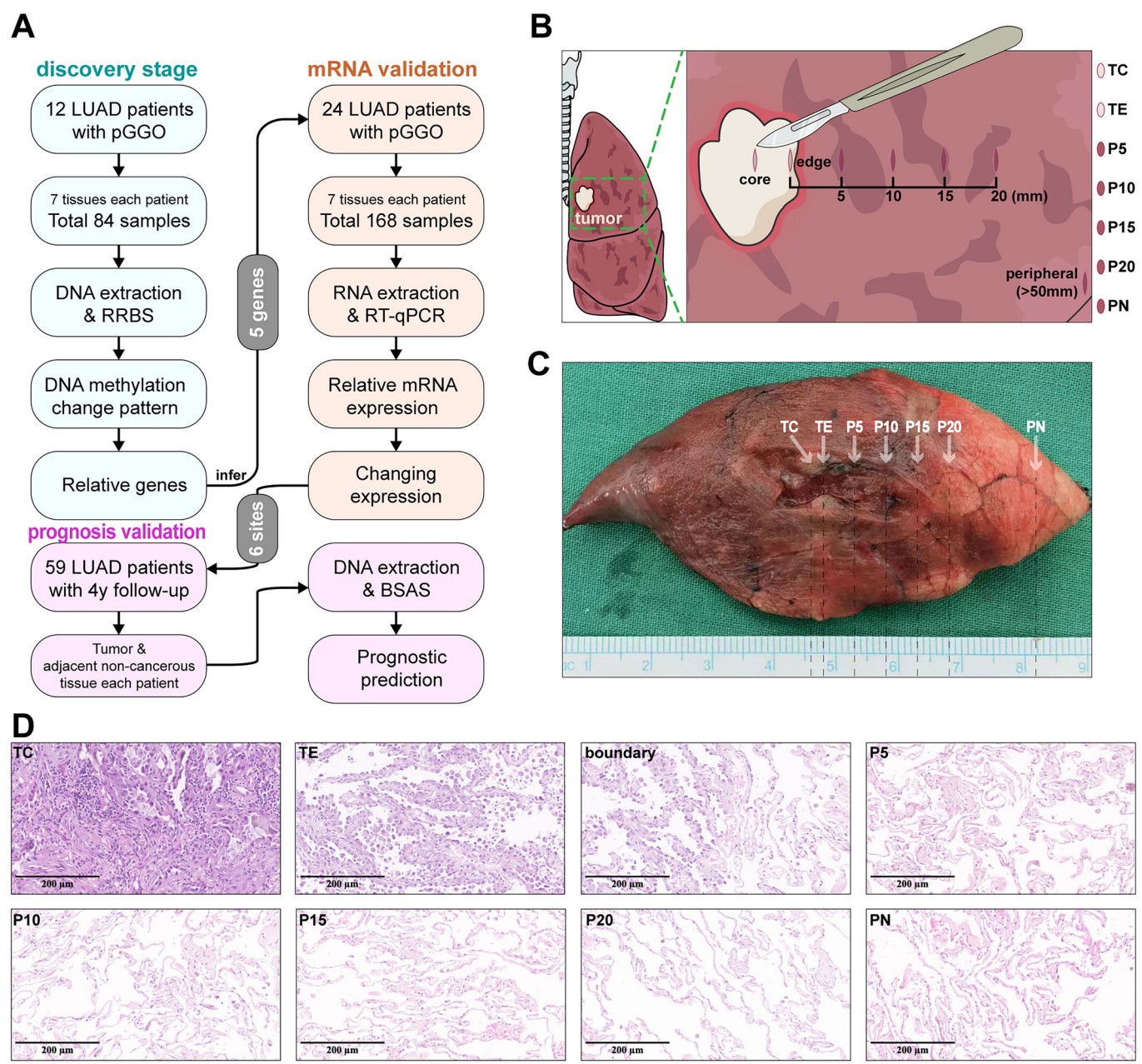

**Figure 1. Design of the sample set and experiment and quality control of methylation data.**

(A) Study design and workflow. The study consisted of three distinct stages: the discovery stage, where early-stage LUAD patients were recruited, and when samples from seven regions were collected postsurgery. DNA was extracted and subjected to RRBS for DNA methylation analysis, and the relevant genes were screened for validation. The mRNA validation stage involved RNA extraction from additional LUAD patients to verify the expression levels of five selected genes via RT-qPCR. The prognosis validation stage included the analysis of 59 LUAD patients with a minimum follow-up period of 4 years. Tumors and adjacent noncancerous tissues were collected from FFPE blocks, and DNA was extracted for BSAS to assess prognostic predictions based on the 6 differential methylation sites. (B) Sampling schematic. Samples were obtained from multiple regions, including the core (TC) and edge (TE) of the tumor, paired-adjacent areas (P5, P10, P15, P20), and a paired-distal normal region (PN) relative to the tumor. (C) A surgically excised sample from the study demonstrates the core and edge of the tumor (TC and TE), paired-adjacent areas (P5, P10, P15, P20), and the paired-distal normal (PN) region. (D) Representative H&E-stained section of a set of samples. Clear morphological differences are observed between the tumor region and paired-adjacent areas (P5, P10, P15, P20), as well as the paired-distal normal region (PN). The tumor boundary is visibly distinguishable at the interface between the tumor and adjacent tissues.

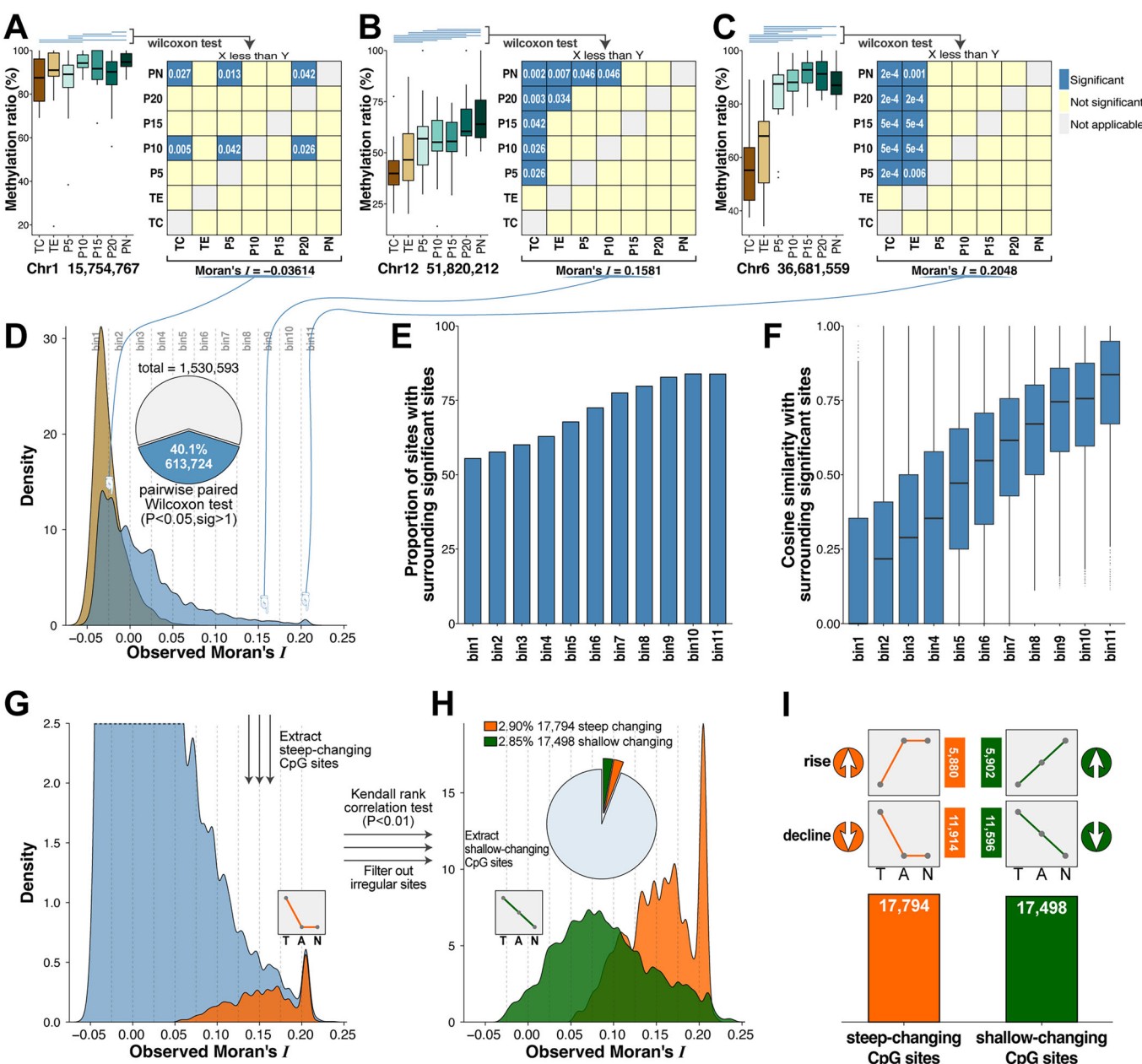

**Figure 2. Analysis of methylation differences between locations and screening of consistent change CpG patterns.**

The boxplots on the left side of (**A–C**) illustrate the methylation levels of CpG sites at seven locations across three representative examples, with the median as the center, box bounds as the 25th and 75th percentiles, and whiskers extending to the minimum and maximum values within 1.5×IQR (n = 12). Outliers beyond 1.5×IQR are plotted as individual points. The blue lines indicate statistically significant pairwise comparisons (one-tailed paired Wilcoxon test, P < 0.05). The 7 × 7 matrix on the right displays the significance of all pairwise combinations. The blue cells represent significant combinations, whereas the beige cells represent nonsignificant combinations. The Moran's I for the matrix is calculated and displayed at the bottom. (**D**) Density plot of Moran's I calculated for the 613,724 CpG sites showing significance in at least two pairwise comparisons across all seven regions. The blue curve represents the observed distribution of Moran's I, whereas the brown curve depicts the distribution of Moran's I from randomly shuffled matrices of these CpG sites. The pie chart shows the proportion of CpG sites meeting the criteria out of all tested CpG sites. CpG sites with Moran's I values < −0.025 and >0.2 were used as the start and end bins, and the range from −0.025 to 0.2 was divided into 11 equal-width bins (each with a width of 0.025). (**E**) For each bin, the proportion of other sites with significant methylation differences within a range of 30 bp upstream and downstream of the bin's sites was calculated. (**F**) For each site with significant methylation differences that neighbor significant sites within 30 bp, the cosine similarity between the matrices of paired sites was calculated and presented in a boxplot. (**G**) The distributions of Moran's I values of steep-changing CpG sites (orange), extracted by Methods, and the proportions of the overall distribution (blue, zoomed) are shown. (**H**) The distribution of Moran's I values of shallow-changing CpG sites (green), extracted via methods, and the comparison with the steep-changing sites (orange). The pie chart shows the proportion of steep- and shallow-changing sites among the total 613,724 CpG sites. (**I**) A summary of the four methylation change patterns and their respective CpG site counts.

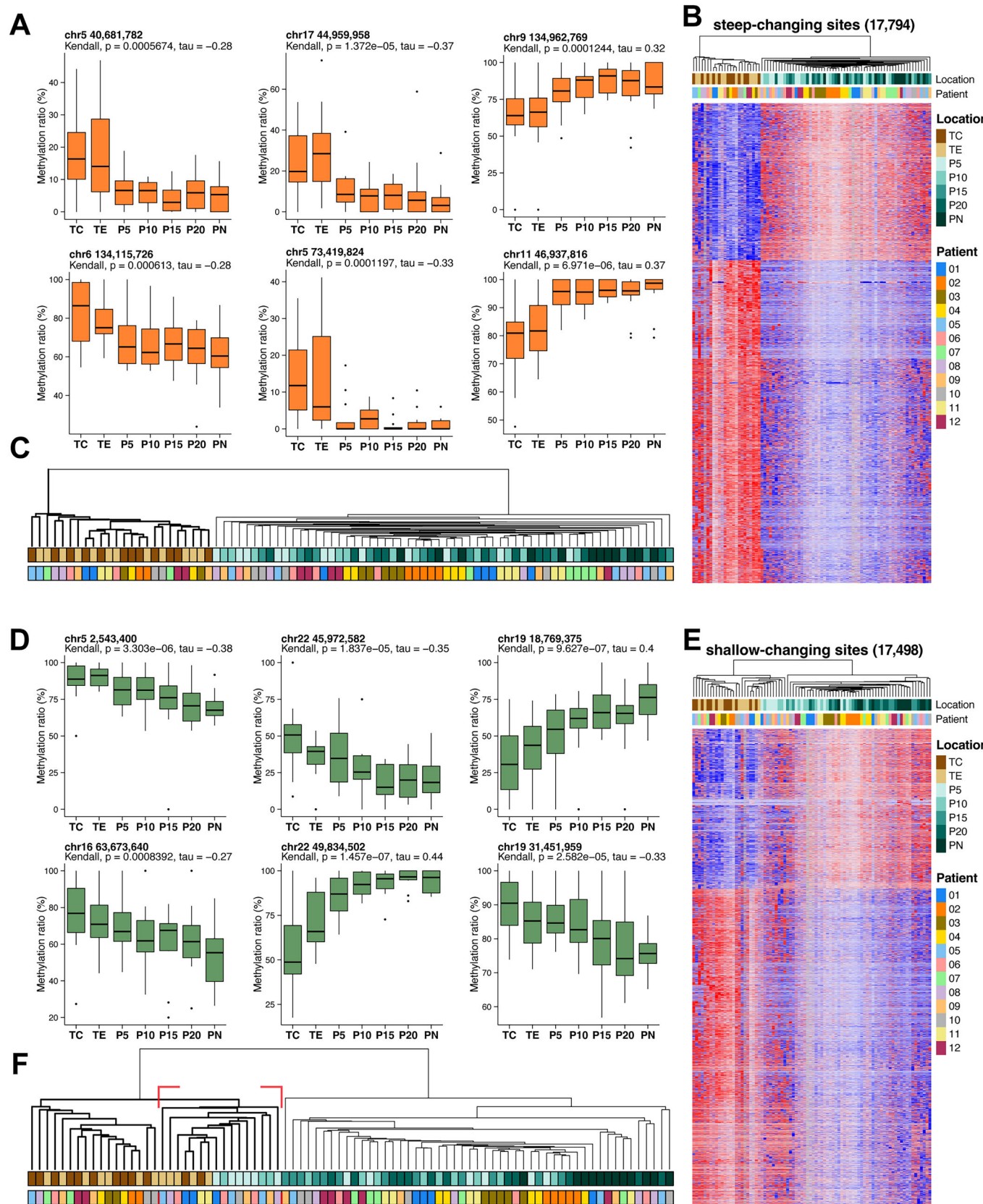

Figure 3.    Feature clustering and partial visualization of T-N changing CpG sites.

(A) Boxplots depicting the methylation levels at various locations for six steep-changing CpG sites, with the median as the center, box bounds as the 25th and 75th percentiles, and whiskers extending to the minimum and maximum values within 1.5×IQR ($n = 12$). Outliers beyond 1.5×IQR are plotted as individual points. The Kendall correlation test was performed, with tau representing the Kendall rank correlation coefficient. (B) Heatmap illustrating the methylation levels of steep-changing CpG sites ($n = 17,794$, rows) across samples ($n = 84$, columns). The data were clustered using Ward.D2 method, and standardized DNA methylation values (Z-scores, row-scaled) were used to visualize the relative methylation differences. (C) The dendrogram resulting from clustering 17,794 steep-changing CpG sites and the corresponding annotation details are shown. The top row indicates different locations, whereas the bottom row represents various patient sources. (D) Boxplots illustrating the DNA methylation levels of six shallow-changing CpG sites across different locations, following the same format as in (A). (E) Heatmap depicting the DNA methylation levels of all 17,498 shallow-changing CpG sites across 84 samples, following the same methods as in (B). (F) The dendrogram resulting from clustering 17,498 shallow-changing CpG sites and the corresponding annotation details are displayed. The red box highlights a branch displaying a mixed distribution of tumor and adjacent normal tissues.

Moran's $I$ values calculated from the actual tissue samples were significantly higher than the simulated random samples, indicating a more ordered pattern among the identified CpG sites.

To further explore this, we divided the CpG sites into 11 bins based on their Moran's $I$ values, using an interval of 0.025, and calculated the proportion of sites that had at least one neighboring site with an ordered pattern within 30 nucleotides upstream or downstream. We observed a positive correlation between Moran's $I$ and this proportion (Fig. 2E). When examining the similarity of the sites, we found that sites with higher Moran's $I$ values exhibited greater similarity among intergroup sites (Fig. 2F). Two example sites were shown in Fig. EV1: at chr2:42905798, three differentially methylated CpG sites were located within its window, two of which were extremely similar to its methylation status pattern; In another example, there were no other differentially methylated sites in the window before and after chr1 1,138,200. Such a trend persisted also under both finer and coarser binning strategies (Appendix Fig. S2), which suggests that the ordered patterns identified were stably present not only in scattered CpG sites but also in specific regions.

To further explore the observed pattern of change, we initially extracted 17,794 CpG sites that exhibited a steep change at the visual boundary of the tumor (between TE and P5), based on the significance of the difference of TC and TE samples compared to noncancerous samples (steep changing sites, shown in orange in Fig. 2G). The remaining sites were then subjected to Kendall's nonparametric test ($P < 0.01$, Fig. EV1B), and sites exhibiting fluctuating changes and insufficient differences between noncancerous samples were excluded. Consequently, we obtained 17,498 sites exhibiting shallow changes in methylation levels across the visual boundary (shallow changing sites, shown in green in Fig. 2H). A detailed description of the analysis is provided in "Methods", Fig. EV2 and Appendix Methods. As a result, we identified two types of sites that effectively reflect the pattern of methylation changes in tumor-adjacent and distal tissues. Among these, 5880 sites exhibited a steep rise trend, 11,914 sites showed a steep decline trend, 5902 sites showed a shallow rise trend, and 11,596 sites showed a shallow decline trend (Fig. 2I; Dataset EV1).

We further compared the genomic distribution characteristics of CpG sites exhibiting steep and shallow methylation changes. Using all other detected CpG sites ($n = 1,495,301$) as a reference, we examined the distribution proportions of steep ($n = 17,794$) and shallow ($n = 17,498$) changing sites within gene bodies, promoters (defined as 2 kb upstream of the transcription start site), CpG islands, transcription factor (TF) binding regions, and CCCTC-binding factor (CTCF) binding regions (Appendix Fig. S3A). Compared with other CpGs, steep/shallow changing sites are less likely to be located in promoter regions, while both were relatively

enriched in TF binding regions. We further stratified the changing sites by rise and decline trends (Appendix Fig. S3B,C), and found that both steep and shallow rise sites were less frequently located in CpG islands, while steep decline sites were more commonly located within these regions. In TF binding regions, gain-of-methylation events were more frequently observed, with 2.30% of steep rise and 1.97% of shallow rise sites located in these regions, several times higher than other CpG sites, and the proportion of decline sites was also higher than that of other CpG sites. These findings suggest that such methylation alterations tend to occur in regions involved in transcriptional regulation, where increased methylation may interfere with transcription factor binding, leading to gene downregulation.

## Steep- and shallow-changing trends demonstrated different boundaries of tumor

To further investigate the relationship between these methylation sites and tumor boundaries, we analyzed the steep and shallow changing sites across visual boundaries separately.

Here, we present the status of six sites as examples. Among the steep-changing sites across regions, we observed that the methylation level of these sites displayed high similarity on each side of the visual boundary of the tumor and a significant increase or decrease from TE to P5 (Fig. 3A). Using the methylation levels of all 17,794 steep-changing CpG sites for clustering analysis of the 84 samples, we observed clear distinctions between the tumor and noncancerous regions (Fig. 3B). Our clustering analysis revealed two distinct branches: one representing the tumor regions and the other representing the noncancerous regions, as shown in Fig. 3C. These findings suggest that the methylation status of these sites is closely associated with tumor development, but is primarily confined to the interior of the tumor tissue.

Conversely, we showed the status of six additional sites exhibiting distinct patterns. The boundary of shallow changing sites across regions was relatively unclear, with the methylation level of the P5-P20 region remaining intermediate between those of the tumor and PN (Fig. 3D). Using the methylation status of these 17,498 sites as the basis for clustering analysis, a group of samples exhibited methylation levels between those of the tumor and PN (Fig. 3E). During clustering, most non-tumor samples remained distinct from tumor tissue, although some P5 and P10 samples clustered alongside the tumor tissues (Fig. 3F). This result not only indicates the methylation of these sites is related to tumor development but also suggests that adjacent tissues within 10 mm of the tumor boundary may no longer be considered strictly normal when evaluated from the perspective of overall methylation differences.

## Steep and shallow changing sites formed regions to regulate genes

DNA methylation changes rarely act through isolated CpG sites; instead, they frequently occur in clusters, forming regionally coordinated methylation patterns—especially within CpG islands —that can have collective effects on gene regulation.

To capture these regional dynamics, we further identified potential regulatory regions among the aforementioned sites and used the median of all ordered pattern CpG sites within each region to represent the overall methylation level of that region. A total of 3021 discrete methylation-changing regions were aggregated based on previously established criteria (Roadmap Epigenomics et al, 2015) (see "Methods"). These regions were then categorized into six methylation change patterns: regions exhibiting steep rise ($n = 245$), steep decline ($n = 510$), shallow rise ($n = 107$), shallow decline ($n = 255$), and regions containing both steep and shallow rise (mixed rise, $n = 441$) or decline (mixed decline, $n = 1463$) sites (Fig. 4A; Dataset EV2). Only 16 CpG regions (0.5%) exhibited both rising and declining CpG sites, indicating that the majority of sites with methylation changing trends were consistent with the overall methylation status within their respective regions; thus, these regions were not included in the subsequent exploration.

The methylation-changing regions were categorized into steep-changing, mixed-changing, and shallow-changing groups, and the 84 samples from the discovery stage were re-clustered based on regional methylation levels (Fig. 4B). The results for steep-changing regions were consistent with those for steep-changing sites, whereas the results for shallow-changing regions were similar to those for shallow-changing sites. Regarding the mixed-changing regions, P5 from two patients clustered with tumors due to their similarity. We considered both shallow-changing and mixed-changing regions as tumor-affected regions and further localized them to gene bodies or promoters of relevant genes (Fig. 4A). Among steep rise regions, 208 genes had gene body overlaps and 82 had promoter overlaps. Among steep decline regions, 467 genes were affected in their gene bodies and 209 in their promoters. For shallow-rise regions, 94 genes were affected in gene bodies and 42 in promoters, while shallow-decline regions overlapped 233 gene bodies and 106 promoters. In the mixed rise group, 385 genes had gene body overlaps and 153 had promoter overlaps; in the mixed decline group, 1168 gene bodies and 588 promoters were affected (Dataset EV2).

Gene Ontology (GO) and Kyoto Encyclopedia of Genes and Genomes (KEGG) analyses were performed on these genes (Fig. 4C; Appendix Fig. S4). Among them, pathways such as "pathways in cancer", "transcriptional misregulation in cancer", "proteoglycans in cancer", and "microRNAs in cancer", along with various pathways associated with carcinogenesis, were identified, suggesting that tumor tissues may alter the status of adjacent tissues through these pathways. Additionally, numerous signaling pathways, including focal adhesion, ECM–receptor interaction, and axon guidance, suggested the shedding, proliferation, and migration of tumor cells (Amit et al, 2016; Bapat et al, 2011). To further illustrate these findings, we examined the cancer-associated gene *WNT7B* as an example. Three regions with T-N declined methylation were identified in its gene body, including nine shallow-decline sites and two steep-decline sites (chr22:45,972,123–45,972,224), as well as four shallow-decline sites

and four steep-decline sites (chr22:45,972,544–45,972,616) located in the first exon (Fig. 4D). The median methylation levels of the two regions' CpG sites are presented in Fig. 4E,F. In the TCGA database, the expression of this gene was significantly upregulated in lung adenocarcinoma tumors (Fig. 4G; Appendix Fig. S5). Aberrant hypermethylation of cancer transcriptional regions leads to overexpression of related genes (Arechederra et al, 2018; Su et al, 2018), which could explain the elevated expression of *WNT7B* in lung adenocarcinoma and suggests that this gene is most likely also upregulated in adjacent tissues surrounding the tumor.

## Relative gene and protein expression confirmed the tumor effect on adjacent tissues

To investigate whether the observed methylation changes across visual boundaries correlated with changes in gene expression, we conducted a validation stage using 24 surgical patient samples of LUAD. In this stage, we sampled these 24 patients from the same seven positions (TC, TE, P5, P10, P15, P20, PN) using the same methodology as before, measuring the relative expression of target mRNAs using RT-qPCR (detailed in "Methods" and Table EV2). Gene expression values were normalized by the expression of the *ACTB* gene, selected as the reference after evaluating multiple candidate housekeeping genes in the TCGA-LUAD cohort. For target selection, we identified five genes among the aforementioned relatively concentrated changing regions and sites, including *CDKN2A*, *FZD10*, *NOTCH1*, *PDGFRA*, and *WNT7B* (Appendix Fig. S5). Among these, *CDKN2A*, *NOTCH1*, and *WNT7B* exhibited both shallow/mixed-changing regions and changing sites, while *FZD10* and *PDGFRA* contained multiple shallow-changing sites.

In total, there were 120 mRNA trend results for each of the five genes across the 24-patient groups, which were classified into three categories: steep changing, shallow changing, and no ordered pattern (Fig. EV3; Dataset EV3). Of these, 38 exhibited steep changing trends, 45 showed shallow changing trends, and the remaining 37 showed no significant changes or ordered patterns (Fig. 5A). To illustrate shallow trends, we present several examples: *CDKN2A* exhibited high expression in TC/TE of Patients 10 and 20, with a shallow decline observed away from the tumors. Similar changes were noted for *WNT7B* in Patients 4 and 19. Conversely, *FZD10* exhibited relatively low expression in TC/TE in Patients 13 and 18, progressively increasing in the direction away from the tumor. Similar changes were noted for *NOTCH1* in Patients 8 and 11, as well as for *PDGFRA* in Patients 2 and 13 (Fig. 5B). Thus, this shallow-changing trend was also reflected in the mRNA expression levels of important genes, suggesting that the tissues beyond the visual boundary of the tumor tissue were influenced by the tumor tissue.

From each patient's perspective, shallow-changing trends of mRNA across visual boundaries were not confined to a single gene but could be simultaneously present in several or even most of these five genes. Conversely, shallow-changing trends of mRNA for each gene were observed in most samples. These findings indicate that the influence of tumor tissue on adjacent tissues could be extensive and warrants further attention and investigation. Furthermore, when combined with other clinical indicators, the impact on tumor-adjacent tissues did not correlate with the size or stage of the tumor, suggesting that this effect may assess tumor status from a novel independent dimension (Fig. 5A).

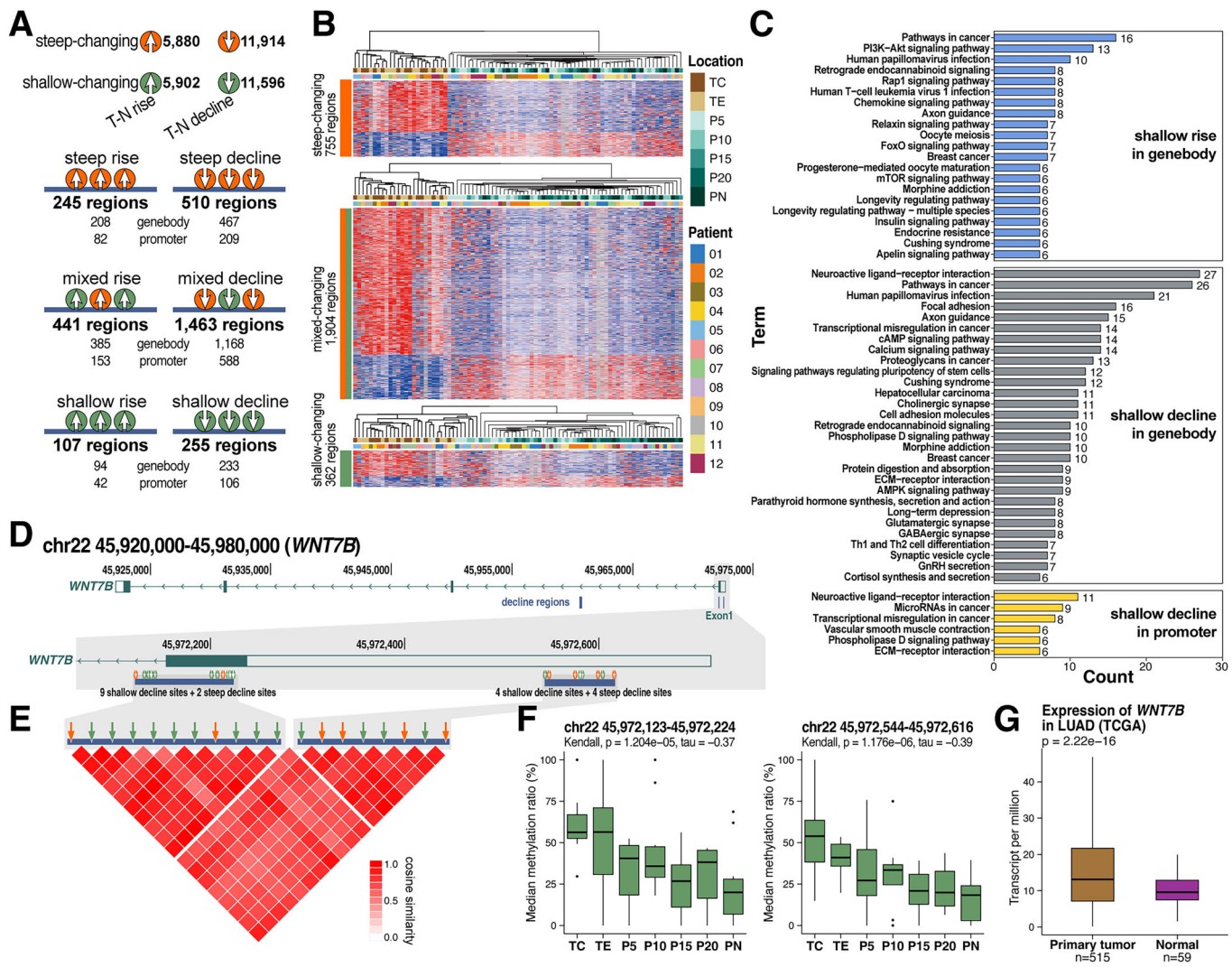

**Figure 4.  Sites in the T-N changing regions showed a consistent change pattern and regulated the expression of related genes.**

(A) The top panel presents the counts of four DNA methylation change types: steep rise/decline and shallow rise/decline. The bottom panel illustrates six DNA methylation region types identified by CpG sites: steep rise, steep decline, shallow rise, and shallow decline regions contain only corresponding changing sites, while mixed rising and mixed declining regions contain both steep- and shallow-changing CpG sites in the same direction. The number of genes regulated by these regions is also indicated. The gene body spans from the transcription start site (TSS) to the transcription end, and promoters are 2 kb upstream of TSS. (B) Three heatmaps displaying median methylation levels for steep-, mixed-, and shallow-changing regions across 84 samples. The methylation level for each region was calculated as the median of CpG site methylation levels within that region. The data were clustered using the Ward.D2 method, and standardized DNA methylation (Z-scores, row-scaled) were applied to visualize the relative methylation differences. (C) KEGG analysis of genes with shallow- or mixed-changing regions, filtered by gene count >5 and P value < 0.05. Bars represent relative gene counts, with genes containing multiple changing regions considered once per change type. (D) The distribution of three decline regions in WNT7B gene body (chr22:45,920,000 to 45,980,000) serves as an example, with the gray box highlighting the first exon containing two changing regions and their CpG sites. (E) Cosine similarities of CpG sites in the two upper regions. (F) DNA methylation levels of the two regions across seven locations (n = 12). The methylation level of each region was calculated by the median of all CpG sites in the region. (G) WNT7B expression levels in primary LUAD and normal tissues were obtained from TCGA. The Mann–Whitney U test is used, and P values are shown. Boxplots present the median as the center, box bounds as the 25th and 75th percentiles, and whiskers extending to the minimum and maximum values within 1.5×IQR. Outliers beyond 1.5×IQR are plotted as individual points.

Building on our methylation and mRNA findings, we next assessed WNT7B protein expression in tumor and adjacent tissues from Patients 4, 15, and 19. FFPE sections encompassing the tumor core (TC), tumor edge (TE), and adjacent tissue (paired-adjacent 0–5 mm; P5) were retrieved and first confirmed by HE staining. Sections were then stained with an anti-WNT7B antibody, and DAB intensity was measured in regions of interest (ROIs) placed at TC, TE, P5 Near, and P5 Far for each sample, as described in "Methods". To mitigate

sampling bias, ROIs were selected along three radial directions per section. Taking Sample 19 as an example (Fig. 5C), we selected three independent sampling directions and selected the corresponding tumor boundary positions to observe in magnification (Fig. 5D). Combining the HE-stained sections at the same position, we can distinguish relatively clear tumor's visual boundaries. In Sample 19, mean DAB OD declined progressively from TC through P5 Far in all three directions, with minimal difference between TE and P5 Near in

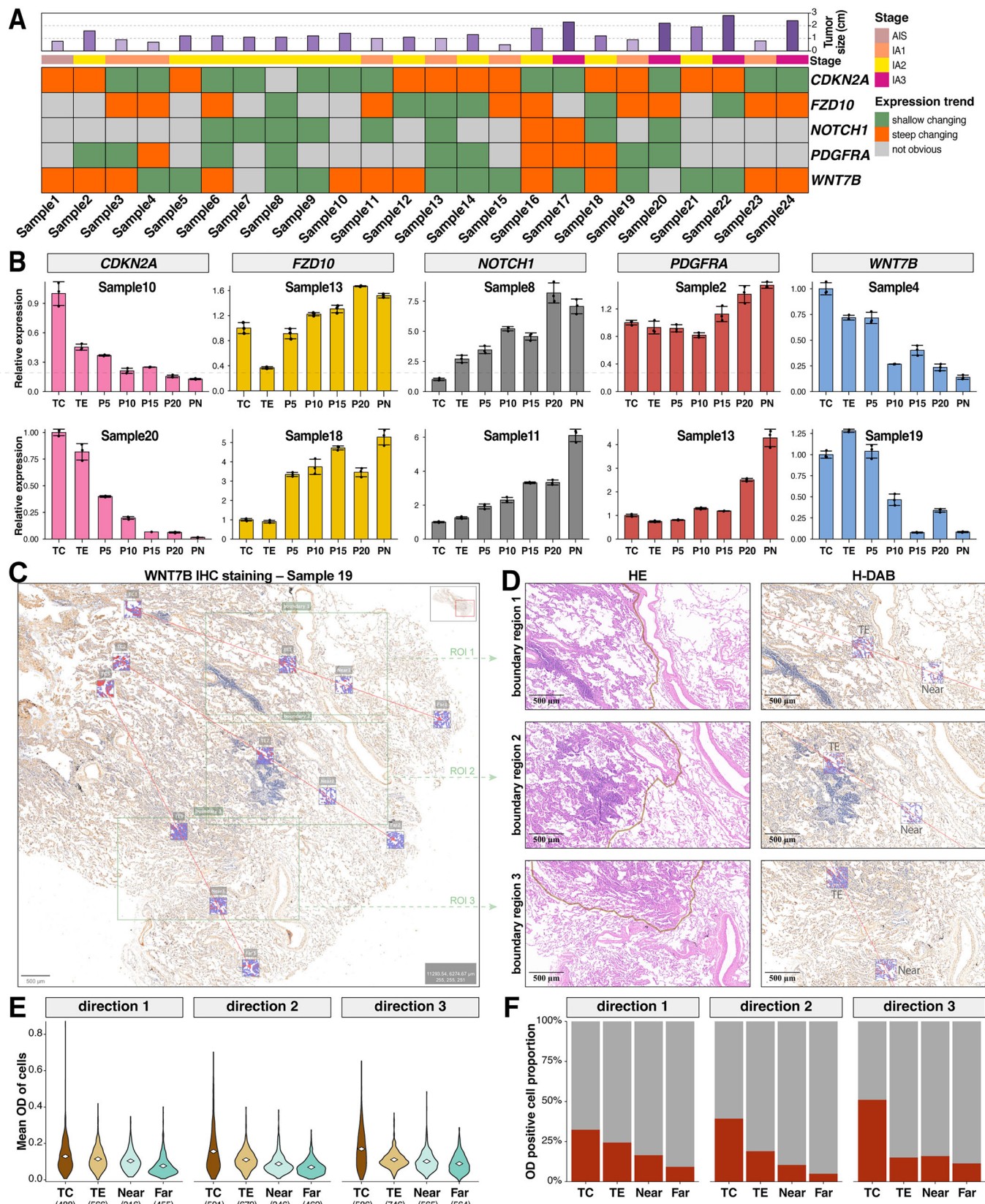

◄

**Figure 5. mRNA and protein expression changes of specific methylation-change genes at different locations.**

(A) Heatmap illustrating the expression trends of selected genes across samples. The top two annotations represent tumor size and TNM stage. (B) Examples of mRNA expression with shallow-changing patterns are presented. Two samples were chosen for each gene. The relative expression levels of each gene were normalized using the reference gene *ACTB* and then standardized to the relative expression of TC for comparison, with error bars indicating the mean ± standard deviation (SD) from three replicates. (C) Representative IHC staining of WNT7B protein in LUAD Sample 19. FFPE tumor tissue section was stained using the WNT7B antibody. Three directions from the tumor core to adjacent tissues were marked by red lines, and 300-μm-squared regions of interest (ROIs) of the tumor core (TC), tumor edge (TE), and adjacent tissue at different distances (P5–Near and P5–Far) in each direction were selected for optical density (OD) analysis. (D) Three larger ROIs (ROI 1, ROI 2, and ROI 3) covering the corresponding TE and P5-Near regions were selected for local magnification display of the three boundary regions in Sample 19. HE and H-DAB staining images showed the visual boundary of the tumor and corresponding DAB signals in TE and P5-Near regions. (E) Violin plots showing the distribution of mean OD per cell for each region (TC, TE, Near, and Far) in three directions of Sample 19, and cell numbers of ROIs were shown below. (F) Proportion of OD-positive cells in each region and direction, based on predefined OD thresholds described in Methods.

direction 3 (Fig. 5E). The proportion of WNT7B-positive cells— defined by an OD threshold of 0.15—mirrored this pattern (Fig. 5F). In Sample 4, P5 Near exhibited higher OD than TE in direction 2, whereas direction 1 showed a clear TC-to-P5 gradient and direction 3 yielded similar OD across all regions (Fig. EV4A–D). For Sample 15, three divergent directions were selected to capture maximal heterogeneity: direction 1 and 3 showed a consistent TC-to-P5 decline, while direction 2 revealed elevated P5 Near expression resembling TE (Fig. EV4E–H). Although the sample size is limited and this analytical strategy cannot fully capture the overall tumor landscape, the protein expression pattern of WNT7B resembles the trends observed in DNA methylation and mRNA levels—namely, in certain tumors or along specific directions within a tumor, no significant changes in protein expression are detected at the tumor boundary.

## Association between the DNA-methylation-changing feature and LUAD patient OS

To validate this phenomenon, we collected an additional 59 lung adenocarcinoma samples, all prepared between March 2016 and April 2020, each with over four years of follow-up data (Table EV1). Among these samples, 18 experienced endpoint events within at least 4 years (additional information presented in Fig. 6A). We selected 6 CpG sites located on the *FZD10*, *NOTCH1*, and *WNT7B* genes as our study subjects: chr9:136,511,336, chr9:136,511,366, chr9:136,511,379, chr12:130,162,026, chr12:130,162,056, and chr22:45,972,580. These sites have been confirmed to exhibit a shallow change phenomenon across tumor visual boundaries in multiple samples, as previously shown. Considering future clinical applications, the comprehensive and precise acquisition of tumor samples and multiple paired-adjacent tissues for comparative analysis is overly complex and costly. Therefore, we propose to simplify the approach by selecting tumor regions and adjacent noncancerous areas from pathology slides for sampling and comparison (Fig. 6B). To obtain DNA methylation information, we employed Bisulfite amplicon sequencing (BSAS) for these 6 sites, which further reduces detection costs than RRBS, as detailed in "Methods" (Table EV2 and Dataset EV3). Due to the simplification of the detection method, the criteria for determining the presence of shallow change phenomena across tumor visual boundaries have also been revised. In brief, we will assess the relative difference in methylation levels between the tumor and adjacent noncancerous regions for each target CpG site. If the difference is below a certain threshold, we will classify it as a low-difference (similar to "shallow") change; otherwise, it will be considered a high-difference change (similar to "steep").

We established a methylation difference threshold range of 3% to 15% and systematically evaluated the hazard ratio (HR) levels for sample stratification based on these six CpG sites across various thresholds (Fig. 6C). The results demonstrate that these sites effectively stratified the 59 samples and revealed significant prognostic differences across most threshold values, with predictive ability gradually declining as the threshold increased. Notably, five sites on *FZD10* and *NOTCH1* exhibited high HRs at lower thresholds. Their HRs decreased as the threshold rose from 3% to 7%, with ranges as follows: chr12:130,162,026 (HR: 1.6–3.6), chr12:130,162,056 (HR: 2.6–3.7), chr9:136,511,336 (HR: 4.3–17), chr9:136,511,366 (HR: 2.1–4.3), and chr9:136,511,379 (HR: 1.6–3.7), as detailed in Fig. EV5. Conversely, the site on WNT7B (chr22:45,972,580) showed an opposite trend, with the low-difference group potentially associated with better survival outcomes, exhibiting HRs ranging from 0 to 0.49 as the threshold increased from 3% to 7%. As wider thresholds may misclassify some "steep change" samples as "shallow change," leading to reduced HRs, the loss of predictive performance is expected.

To account for variability in methylation sequencing, we selected a 5% threshold for further analysis. Under this condition, the sites chr12:130,162,026 and chr12:130,162,056 (*FZD10*) and chr9:136,511,336, chr9:136,511,366, and chr9:136,511,379 (*NOTCH1*) demonstrated prognostic risk stratification capabilities, with HRs of 1.6, 2.6, 7.3, 2.1, and 2.8, respectively (Fig. 6D–I). Using these five sites on *FZD10* and *NOTCH1* as criteria, the high-difference change group exhibited a significantly better prognosis than the low-difference (shallow) change group, indicating a negative impact of the tumor on surrounding tissue.

Given that many patients do not exhibit completely consistent trends across multiple sites, and to enhance the model's robustness, we aimed to further evaluate the performance of a combined model that incorporates these sites. Among five sites exhibiting similar trends (chr9:136,511,336, chr9:136,511,366, chr9:136,511,379, chr12:130,162,026, and chr12:130,162,056), if at least N sites meet the criteria for shallow change between the tumor and adjacent tissues, the patient is classified into the high-risk group; otherwise, the patient is categorized as low-risk (Fig. 6J). When $N = 2$, the results from the Kaplan–Meier survival curves reveal a significant prognostic difference between the high-risk and low-risk groups, with an HR of 4.9 (Fig. 6K, more results are shown in Appendix Fig. S6). In addition, we examined the TNM staging in the risk groups derived from the combined models ($N = 2$) (Fig. 6L). Notably, the majority of patients in stages III and IV were classified into the high-risk group, with only one stage III patient in the low-risk group; all other low-risk patients were in stage I. Furthermore,

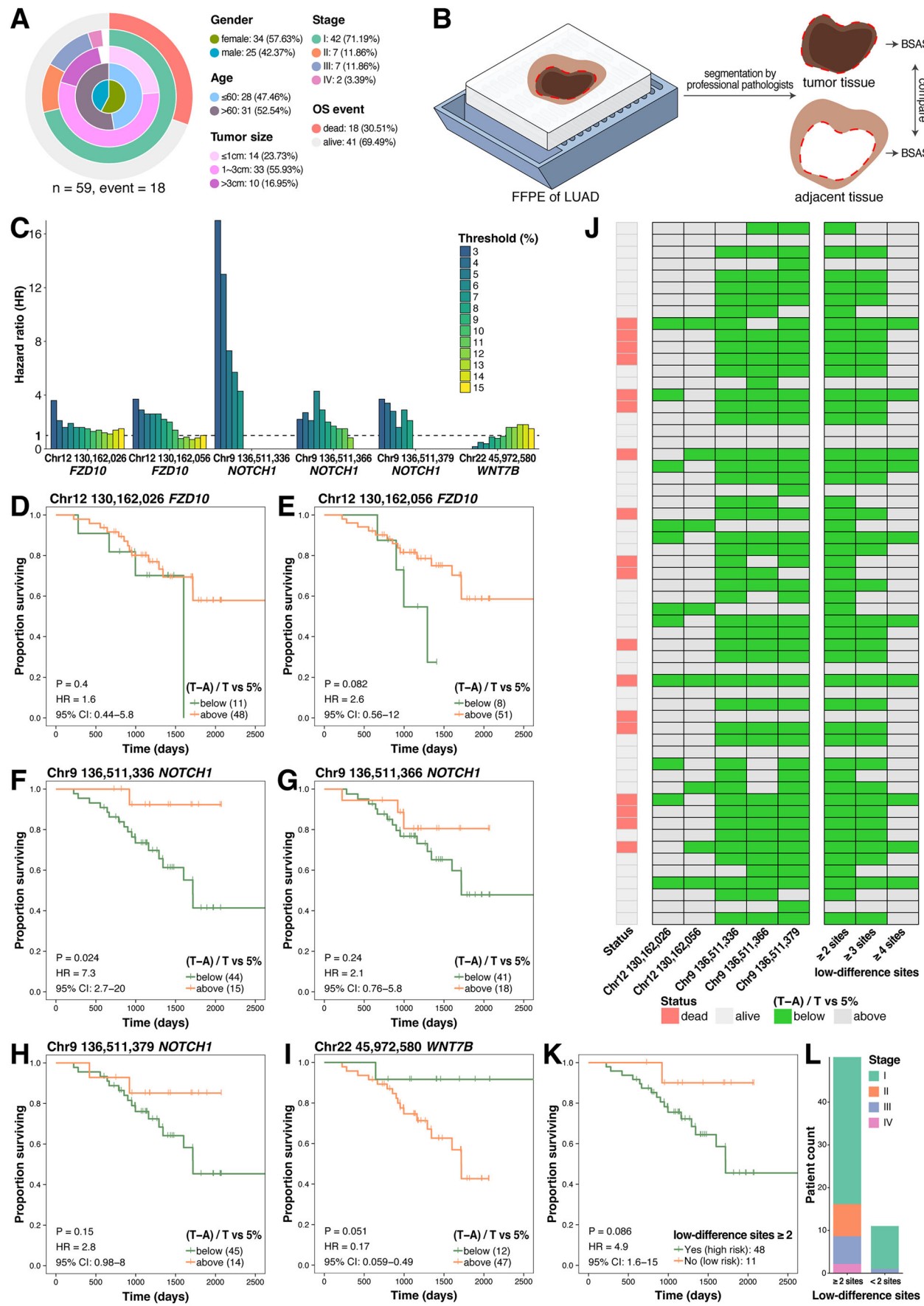

**Figure 6. DNA methylation of shallow change sites in the 59 validation cohort and their relationship with patient prognosis.**

(A) Demographic and clinical characteristics of the 59 LUAD patients in the prognosis validation cohort are presented. Patients are categorized by sex, age, tumor size, stage, and survival status. (B) A schematic representation of sample processing is shown. Formalin-fixed, paraffin-embedded (FFPE) LUAD tissue was segmented by pathologists into tumor and adjacent noncancerous regions, followed by BSAS analysis. (C) Risk stratification performance with different ratios as thresholds at each site. (D–I) Kaplan–Meier survival curves illustrate the relationship between shallow DNA methylation changes at individual CpG sites and overall survival in patients with a 5% methylation relative difference as a threshold. The hazard ratios (HRs) and P values for survival differences between groups with steep and shallow methylation changes are indicated. The genes examined included *FZD10*, *NOTCH1*, and *WNT7B* across various chromosomes and methylation sites. (J) Heatmap displaying the DNA methylation change status at the six selected shallow change sites for each patient. Patients are color-coded by survival status (green = alive, red = dead) and methylation difference (gray = high difference, green = low difference). The right panel shows the cumulative number of patients with shallow changes at one, two, or more sites. (K) K–M survival analysis for patients with shallow DNA methylation changes at two or more sites. The survival difference between patients at high risk (≥2 sites with shallow change) and those at low risk (<2 sites) is shown. (L) Bar plot illustrating the distribution of patients in high- and low-risk groups based on shallow changes across various TNM stages. P values of K–M survival analysis were calculated using the chi-squared distribution.

the high-risk classification for stage I and II patients corresponded with a higher proportion of endpoint events and poorer prognosis, demonstrating that this model can effectively predict risk in early-stage lung adenocarcinoma patients, highlighting its significant clinical value.

## Discussion

In this study, we aimed to investigate the impact of tumors on adjacent tissues beyond the visible boundary by utilizing surgical tissues from patients with LUAD. In the discovery stage, we analyzed samples from 12 patients and observed approximately 17,000 sites exhibiting a steep change in methylation levels at the tumor's visual boundary, as well as roughly 17,000 sites displaying a shallow change across this boundary. In the mRNA validation stage, which included 24 patients, we found that approximately one-third of the samples exhibited shallow changes in mRNA expression levels for the selected 5 genes across the tumor's visual boundary. These results support the notion that tissues bordering early-stage LUAD tumors are affected and not in a normal state, as confirmed by both DNA methylation, mRNA expression levels, as well as several samples at the protein level. This discovery highlights the widespread influence of tumors on adjacent tissues and their significant role in tumor development, thereby necessitating further investigation. Additionally, we demonstrated through a cohort study that the prognosis of patients can be effectively assessed by evaluating differences in tumors and their adjacent regions based on several methylation sites with changes across the visual border of the tumor, demonstrating the value of this clinical study from a novel perspective.

The study published by Jia et al (2021) revealed that early LUAD tumor tissue affects the methylation status of some sites and regions in surrounding tissues, providing an important foundation for our research. Therefore, we adhered to the sampling criteria established in the previous study and employed the same DNA methylation detection approach to investigate the epigenetic influence of tumors on surrounding tissues. First, the definitions of tumor effect and the criteria for site screening were the most important differences between the two studies. Our study differs significantly from previous research in terms of the amount of data included in the analysis, the richness of analytical methods used, the depth of study, and the questions addressed. Previous research mainly identified differentially methylated regions (DMRs) and CpG sites (DMCs) between TC and PN tissues, but did not fully

capture methylation patterns across the entire tumor-affected area. However, this method may overlook overall methylation trends and abnormalities in many other sites, leading to omissions and misjudgments. In our study, we employed a more systematic and comprehensive approach for site distinction (comparison list provided in Table EV3). For example, the methylation of chr12:5,888,001 significantly differed between tumor and non-tumor tissues; however, the methylation levels in non-tumor tissues fluctuated with distance, showing no stable and continuous changing trend. This was excluded in our analysis but would have been classified as steep in previous studies (Fig. EV2B); There was no significant correlation between the methylation level and the position of chr22:50,029,556, but because the difference between TC and PN and the change in TC-TE-P5 were met, it would have been judged as a shallow-changing site by previous standards (Fig. EV2C); The differences between chr16:991,167 and chr19:22,215,478 in TE-P5 were greater than the TC-TE differences, which would have been judged as steep in previous analyses. However, from an overall perspective, the methylation levels of adjacent positions (P5, P10) changed to resemble the tumor state compared with the normal state (PN), leading to their identification as shallow-changing in our study (Fig. EV2F,G). Furthermore, for the first time, we validated the widespread influence of early-stage LUAD tumor on the expression of important genes in an mRNA validation cohort of 24 patients, along with the prognostic prediction capacity of DNA methylation at these genes in a separate validation cohort of 59 patients. This finding not only confirms the biological significance of DNA methylation changes but also assists us and future researchers in identifying potential targets that may inhibit tumor development and metastasis.

To further evaluate the observation that some adjacent tissues clustered with tumor samples when using shallow changing CpG sites, we conducted hierarchical clustering analyses at multiple levels (see "Methods"). Clustering based on all ~1.53 million CpG sites failed to reveal distinct tumor-normal separation, suggesting that the relevant methylation changes are confined to a small subset of sites (Appendix Fig. S7A). When using the filtered set of ~610,000 CpGs, tumor samples formed more defined clusters, although certain TE samples grouped with adjacent tissues—possibly reflecting individual-specific methylation patterns (Appendix Fig. S7B). In contrast, clustering based on the combined set of steep and shallow changing sites ($n = 35,292$) produced a clearer tumor-normal separation, likely due to the dominant tumor-associated patterns of steep changing sites (Appendix Fig. S7C). The subtle trends carried by shallow changing sites may be masked

when combined with stronger differential patterns, potentially explaining why such patterns have been underrecognized in previous studies. In addition, to robustly assess clustering reliability, we further applied multiscale bootstrap resampling using pvclust, which supported the clustering of some P5/P10 samples with tumor tissues at high bootstrap values based on shallow changing sites, reinforcing the interpretability of these clustering results (Appendix Fig. S7D).

How is DNA methylation affected in the adjacent tissues? Based on the present study, there are at least two possibilities: (1) DNA fragments of tumors may be directly delivered to adjacent tissues. Tumor tissue cells frequently undergo DNA recombination, breakage, and repair, resulting in the formation of numerous DNA fragments (Li et al, 2018b; Rosswog et al, 2021; Shoshani et al, 2021). In addition, extrachromosomal circular DNA (eccDNA) is also produced in tumor cells, which self-replicate and carry both the sequence and methylation information of the tumor tissue (Cohen et al, 2006; Koche et al, 2020; Kumar et al, 2020; Ling et al, 2021; Sin et al, 2020; Wang et al, 2021b). A recent study showed that tumor cells exchange materials, including DNA, through contact with erythrocytes (Liang et al, 2023). Therefore, it is highly likely that adjacent tissue cells, which are in direct contact with tumor tissue, also directly acquire some of the DNA fragments produced by tumor tissue, thus exhibiting a trend of shallow changing in some regions and sites. To confirm this hypothesis, we examined the SNPs present in the patients' methylation sequencing data across each region. However, we did not observe any mutations transferred from TC/TE that could be detected in the adjacent regions of any of the 12 patients. Although this approach is somewhat limited, the preliminary results did not support this hypothesis. (2) Tumor tissues actively or passively may cause alterations in the surrounding microenvironment, leading to changes in methylation and gene expression in adjacent tissues. Previous studies have established that cancer progression and metastasis depend on the interaction between cancer cells and their microenvironment (Amit et al, 2016; Lim et al, 2017; Ombrato et al, 2019), which is influenced by tumor metabolism, signaling pathways, immune cells, angiogenesis, and innervation (Baharom et al, 2022; Cao, 2013; Gajewski et al, 2013; Gysler and Drapkin, 2021; Lei et al, 2020), all of which are important for tumor growth, immune escape, drug resistance acquisition, and migration (Genova et al, 2021; Lim et al, 2020; Sethi et al, 1999; Vitale et al, 2019; Wei et al, 2021). From the GO/KEGG analysis of genes associated with the sites and regions where we observed shallow changes across the visual boundaries of tumors, it is evident that these genes are closely linked to tumor development (Fig. 4C; Appendix Fig. S4). Therefore, this explanation may play a more dominant role and warrants further investigation in future studies.

To explore the cellular basis underlying shallow methylation-associated transcriptional changes, we analyzed single-cell RNA-seq data from 10 paired LUAD and normal lung tissues. We observed upregulation of CDKN2A and WNT7B in tumor tissues, potentially driven by increased proportions of basal epithelial cells, ciliated epithelial cells, and stem-like adenocarcinoma cells (Appendix Fig. S8). In normal tissues, concurrent upregulation of these genes might suggest possible dissemination of stem-like tumor cells or state transitions of normal epithelial cells toward tumor-like phenotypes. This hypothesis is further supported by our IHC results, which showed organized epithelial-like WNT7B

positivity in P5 regions. Additionally, tumor-adjacent tissues contained higher numbers of immune cells—including tumor-associated macrophages, alveolar macrophages, and NK cells—with elevated NOTCH1 expression, indicating potential immune cell recruitment into tumors via chemotactic signals. PDGFRA was highly expressed in fibroblasts in both tumor and normal tissues without significant differences, suggesting patient-specific variability. To dissect cell-type contributions, we compared gene expression after excluding shared cell types within each patient (Appendix Fig. S9). Epithelial cells contributed most to CDKN2A variation, fibroblasts to PDGFRA, and both epithelial and myeloid cells influenced WNT7B expression. Furthermore, clustering based on DNA methylation shallow-changed genes revealed tumor-normal separation predominantly in epithelial and myeloid lineages (Appendix Fig. S10). These findings suggest that shallow epigenetic transitions may reflect gradual regulatory reprogramming in specific cell compartments—potentially related to EMT, immune modulation, or resistance. Notably, activation or reprogramming of normal epithelial cells at tumor-adjacent sites, driven by methylation changes, may represent a key mechanism. Due to the limitations of the third-party dataset, these observations remain exploratory. Future studies integrating spatial transcriptomics, proteomics, and single-cell approaches, guided by our sampling framework, may yield deeper mechanistic insights.

In our mRNA validation stage, we selected five genes to explore whether their expression was affected by tumors in adjacent tissues, such as DNA methylation. This result linked relatively scattered methylation changes upstream to certain gene expressions, with important implications for subsequent studies and the discovery of potential tumor therapeutic targets. CDKN2A is a growth suppressor gene that encodes p16(INK4a)/p14(ARF) and is directly associated with tumor immunity, thought to be a prognostic marker in a variety of cancers (Chen et al, 2021; Romagosa et al, 2011; Tong et al, 2011; Zhao et al, 2016). FZD10 encodes the WNT protein receptor, which belongs to the frizzled gene family and is associated with drug resistance in hepatocellular carcinoma (Wang et al, 2023). This gene is significantly underexpressed in LUAD but has received less attention in lung cancer research. NOTCH1 encodes a group of receptors involved in the Notch signaling pathway, which regulates processes related to cell fate specification, differentiation, proliferation, and survival (Guo et al, 2024; Hori et al, 2013; Koch and Radtke, 2007; Meurette and Mehlen, 2018; Ntziachristos et al, 2014). Researchers have found that this pathway has both tumor-suppressive and pro-cancer effects and generates the tumor microenvironment through its activation in small-cell lung cancer (Lim et al, 2017). PDGFRA encodes a tyrosine kinase receptor involved in the regulation of embryonic development, cell proliferation, survival, and chemotaxis (Bazenet et al, 1996; Elling et al, 2011; Vantler et al, 2006). PDGFRA-expressing cells are recognized fibroblast markers and mesenchymal progenitors (Li et al, 2018a; Muhl et al, 2020), suggesting that methylation changes in its adjacent tissues may relate to tumor microenvironment formation and progression. WNT7B is a clear cancer-related gene, and its expression was significantly upregulated in lung adenocarcinoma tumors in the TCGA database. The WNT family proteins are widely involved in embryonic development and carcinogenesis (Chen et al, 2022; Tennis et al, 2007). WNT7B was recently shown to be significantly correlated with LUSC and OSCC development and was thought to be involved in promoting the

ability of tumor cells to invade the surrounding area (Chen et al, 2022). In addition to the five genes included in our study, we also observed methylation changes at key sites in a lot of genes that have not been previously reported to be associated with lung cancer. For example, PSMA5 is a subunit part of the proteasome, a multi-catalytic proteinase complex with a highly ordered ring-shaped 20S core structure. Another gene, *KIF11*, encodes a motor protein belonging to the kinesin family. Members of this protein family are involved in various spindle dynamics. The functional roles of the gene product include chromosome positioning, centrosome separation, and establishment of the bipolar spindle during cell mitosis. Both genes contained several shallow-rise CpG sites in the promoter regions and were highly expressed in lung cancer according to the TCGA database and significantly associated with poor prognosis. The results of this study will help us to find more target genes that are important for the development of LUAD.

In terms of clinical application, the extent of tumor tissue impact is a crucial reference indicator in oncology diagnosis and treatment. In earlier studies, TNM staging served as an indicator for assessing tumor characteristics, representing the degree of malignancy based on three dimensions: tumor size, lymph node metastasis, and distant metastasis, thereby providing prognostic insight for cancer patients (Carr, 1977). Subsequently, numerous studies have evaluated tumor prognosis from multiple perspectives. Among these, DNA methylation in tumor tissue has been the most extensively studied prognostic biomarker, demonstrating significant potential, with targeted predictive models and therapeutic targets in other cancers aimed at achieving the most accurate assessments (Guo et al, 2019; Gurung et al, 2020; Luo et al, 2020; Ma et al, 2023; Xi et al, 2022; Xiao et al, 2022).

DNA methylation closely correlates with the expression of corresponding genes and reflects the state of cells and tissues. In our study, we indirectly indicated the extent of tumor influence on surrounding tissues through differences in DNA methylation between cells and tissues. Such influences are closely associated with tumor characteristics and patient prognosis. For instance, the influence of genes *NOTCH1* and *FZD10* in surrounding tissues is positively correlated with adverse patient outcomes, suggesting that tumors alter the expression of these genes, providing a competitive advantage during subsequent invasion and metastasis. Conversely, the influence of gene *WNT7B* in surrounding tissues is negatively correlated with poor prognosis, implying that such signals may better attract the attention of the immune system, thereby hindering tumor progression. Due to limitations in cost and time, we selected only 6 CpG sites for this phase of validation. However, we believe that many of the sites exhibiting shallow changes in the aforementioned results may also be closely related to methylation differences between tumor and adjacent noncancerous tissues, warranting further investigation in future studies. Future studies could focus on optimizing detection methods, refining tumor-adjacent tissue comparison strategies, and conducting large-scale clinical trials to build more diverse and representative training cohorts. High-throughput screening and model development will also be essential for improving clinical applicability. While mRNA may serve as a reliable prognostic indicator, limitations in sample availability—particularly those with region-specific preservation, high-quality RNA, and long-term follow-up—currently preclude retrospective validation in LUAD cohorts. In addition, it could also serve as a supplementary perspective for the ongoing controversy over the selection of lobectomy versus segmentectomy for non-small cell lung cancer (NSCLC) (Saji et al, 2022). Furthermore, the expression characteristics of these genes in tumors and surrounding tissues provide new avenues for developing therapeutic drugs.

# Methods

**Reagents and tools table**

| Reagent/resource | Reference or source | Identifier or catalog number |
|---|---|---|
| **Experimental models** | | |
| NA | | |
| **Recombinant DNA** | | |
| NA | | |
| **Antibodies** | | |
| WNT7B primary antibody | Affinity Biosciences | DF9042 |
| **Oligonucleotides and other sequence-based reagents** | | |
| RT-qPCR primer | This study | Table EV2 |
| BSAS primer | This study | Table EV2 |
| **Chemicals, enzymes, and other reagents** | | |
| QIAamp DNA FFPE Tissue Kit | Qiagen | 56404 |
| Methylcode Bisulfite Conversion Kit | *Thermo Fisher* Scientific | MECOV50 |
| TRIzol | *Thermo Fisher* Scientific | 15596026 |
| HiScript II 1st Strand cDNA Synthesis Kit | Vazyme Biotech Co., Ltd. | R212 |
| ChamQ Universal SYBR qPCR Master Mix | Vazyme Biotech Co., Ltd. | Q711 |
| DAB Peroxidase Substrate Kit for IHC | Yeasen Biotechnology (Shanghai) Co., Ltd | 36302ES01 |
| FastPure FFPE DNA Isolation Kit | Vazyme Biotech Co., Ltd. | DC105 |
| EZ DNA Methylation-Gold Kit | Zymo Research, Inc. | D5006 |
| Ex Taq Hot Start Version | Takara Bio Inc. | RR006 |
| New Gel Mini Purification Kit | Zoman Biotech Co., Ltd. | ZPN202 |
| **Software** | | |
| FastQC | https://www.bioinformatics.babraham.ac.uk/projects/fastqc/ | |
| Trim Galore | https://github.com/FelixKrueger/TrimGalore | |
| Bismark | https://github.com/FelixKrueger/Bismark | |
| Mclust | https://mclust-org.github.io | |
| Ape | https://emmanuelparadis.github.io | |
| Rstaix | https://rpkgs.datanovia.com/rstatix/ | |

| Reagent/resource | Reference or source | Identifier or catalog number |
|---|---|---|
| Pvclust | https://github.com/shimo-lab/pvclust | |
| DAVID | https://davidbioinformatics.nih.gov | |
| ComplexHeatmap | https://github.com/jokergoo/ComplexHeatmap | |
| QuPath | https://qupath.github.io | |
| Seurat v5 | https://satijalab.org/seurat | |
| DESeq2 | https://bioconductor.org/packages/release/bioc/html/DESeq2.html | |
| **Other** | | |
| Illumina Hiseq X Ten | Illumina | |
| Illumina NovaSeq 6000 | Illumina | |
| LG-S80 | Wuhan Servicebio Technology Co., Ltd. | |
| CFX 96 | Bio-Rad Laboratories, Hercules, CA, USA | |

## Cohort and sampling

The study enrolled 12 patients with early-stage lung adenocarcinoma (LUAD) exhibiting CT features of pure ground–glass opacity in the discovery stage and 24 patients in the mRNA validation stage. Two radiologists evaluated the CT features and measurements in a back-to-back manner. All patients received regular antibiotic treatment before the diagnosis of early-stage lung cancer. Pathology reports and staging data were reviewed by a certified thoracic cancer pathologist. For the prognosis validation stage, 59 samples were collected between 2016 and 2020. All samples had well-documented follow-up information for more than four years. Ethical approval was granted by the Ethics Committee of Peking Union Medical College Hospital (ZS-1329&JS-2968), and written informed consent was obtained from all participants. In accordance with personal privacy and confidentiality requirements, raw sequencing data were provided upon request.

Tissues were sampled separately from each enrolled patient from various locations, including the tumor core (TC), macroscopic tumor edge (TE), histologically normal tissues adjacent to the tumor at varying distances (P5, P10, P15, P20), and peripheral normal tissue (PN). The samples were stored in formaldehyde for DNA methylation sequencing (discovery stage) and RNA later for RNA extraction (mRNA validation stage) within 30 min after resection. In the prognosis validation stage, samples were prepared into formalin-fixed and paraffin-embedded (FFPE) tissues.

## DNA methylation sequencing and quality control of data

DNA was extracted from stage 1 samples via the QIAamp DNA FFPE Tissue Kit (Qiagen, Hilden, Germany) following the manufacturer's instructions. DNA methylation libraries were prepared via a reduced representation bisulfite sequencing (RRBS) protocol. Briefly, unmethylated cytosines in 200 ng of input DNA

were converted to uracil via the Methylcode Bisulfite Conversion Kit (Thermo Fisher, MECOV50). The converted DNA samples were dephosphorylated and ligated to a universal adapter and then amplified via PCR to add indexes. The libraries were sequenced on an Illumina HiSeq X10 platform.

Sequencing quality was evaluated, and low-quality reads and adapters were removed via Trim Galore (Krueger et al, 2023). Clean reads were aligned to the human reference genome (GRCh38) via Bismark with bowtie2 aligner (Krueger and Andrews, 2011). The methylation ratios of CpG sites were extracted via bismark_methylation_extractor and coverage2cytosine. To ensure high-quality data, sites with less than 15× coverage were filtered, resulting in a total of 4,343,110 CpG sites. The detection frequency distribution of each site in 84 samples is presented in Appendix Fig. S1A. Given the varying detection frequencies of each site across samples and the nature of RRBS, the expectation–maximization (EM) algorithm (Scrucca et al, 2016) was employed for mixture estimation to select sites in the high-frequency distribution for subsequent analysis, retaining 1,530,593 sites detected in at least 68 samples (Appendix Fig. S1B).

## Differences in DNA methylation

We grouped each CpG site by sample location and compared methylation between groups pairwise. The paired one-tailed Wilcoxon test was used to determine the significance with a $P < 0.05$ threshold, and finally, a total of 613,724 CpG sites were obtained with significance between at least two comparisons. On the basis of the results of the group comparisons, a $7 \times 7$ logical matrix can be constructed as the significant matrix, in which each position indicates whether the difference between the two groups is significant. Detailed information regarding data processing, filtering, and classification can be found in Appendix Methods.

## Moran's I analysis

Moran's $I$ is a measure of the overall clustering of the spatial data, showing a correlation in a signal among nearby locations in space, and is calculated for $n$ observations on a variable $x$ at locations $i$, $j$ as

$$I = \frac{\sum_{i=1}^{n} \sum_{j \neq i}^{n} w_{ij} (x_i - \overline{x})(x_j - \overline{x})}{S^2 \sum_{i=1}^{n} \sum_{j \neq i}^{n} w_{ij}},$$

where $x_i$ denotes the observed value at location $i$, $\overline{x}$ is the mean of the $x$ variable over the $n$ locations, $S^2 = \frac{1}{n} \sum_{i=1}^{n} (x_i - \overline{x})^2$, and $w_{ij}$ is the element of the spatial weights matrix for locations $i$ and $j$.(Zhou and Lin, 2008) Here, we used the inverse distance matrix and computed Moran's $I$ autocorrelation coefficient on $7 \times 7$ matrices of 613,724 CpG sites via the R package ape (Paradis and Schliep, 2019). Then, distribution statistics were performed on the corresponding Moran's $I$. In addition, we randomly shuffled the matrices of each site (10 times in a row) to simulate the completely random occurrence of differences between groups and counted the Moran's $I$ distribution of these matrices.

We divided the distribution into 11 bins based on their Moran's $I$ values, with fixed intervals of 0.025 ranging from $\leq -0.025$ to $>0.2$. Each bin thus represents a different level of local spatial

autocorrelation in differential methylation. In addition to the fixed 11-bin scheme based on Moran's I values with intervals of 0.025 (ranging from ≤−0.025 to >0.2), we tested several alternative binning strategies to evaluate the robustness of the classification. These included: (1) finer fixed-width intervals of 0.0125 (21 bins), (2) coarser fixed-width intervals of 0.05 (6 bins), (3) quantile-based binning with 10 equal-sized groups, (4) quantile-based binning with 20 groups, (5) Gaussian mixture model (GMM) binning using the expectation-maximization (EM) algorithm (12 bins), and (6) Fisher–Jenks natural breaks optimization (11 bins), which minimizes within-group variance.

Within each bin, we quantified the proportion of CpG sites that had at least one nearby significantly differential site within a ± 30 nucleotide range. A site was considered to have a nearby significant neighbor if there existed another significant site within this window, two examples are detailed in Fig. EV1C,E, and thus the proportion of sites in each bin could be calculated.

Then, the cosine similarity of the significant logical matrix between these significant difference sites and other adjacent significant difference sites is calculated. We computed the cosine similarity between each site's differential status vector (a 42-dimensional binary vector representing significance status across all location pairs) and those of its neighboring significant sites within ±30 nucleotides.

## DNA methylation changing trend statistics

We filtered the CpG sites showing steep changes between tumor and normal (T-N) according to the significant matrix, requiring that significant differences occurred only between TC and P5-PN, as well as between TE and P5-PN, with no fewer than three significant occurrences for each comparison.

In the order of TC, TE, P5, P10, P15, P20, and PN, we conducted ranking-based nonparametric tests, calculated Kendall's rank correlation coefficient of methylation at each significantly different site, and applied a threshold of $P < 0.01$ for screening. On the basis of the sign of Kendall's rank correlation coefficient τ (tau), we determined the changing trend of methylation from tumor to normal. The analysis was primarily conducted via the R package rstaix.

The sites with nonmonotonic significance, no significance between TC/TE and PN, or no significance between P5-PN in the difference significance matrix were subsequently screened out, and the remaining CpG sites were classified as methylation shallow changed sites.

## Analysis of DNA methylation change sites and related genes

We required that each methylation-changing region contain at least three changed CpG sites exhibiting the same trend, with the distance between two adjacent sites not exceeding 250 bp (Roadmap Epigenomics et al, 2015). The median methylation level of all changing CpG sites represented the region's methylation level. On the basis of the human genome annotation provided by GENCODE (Release 40) (Frankish et al, 2021), we defined the complete transcription region of the gene as the gene body and designated the region 2 kb upstream of the transcription start site the promoter region to map the regions with distance-related methylation changes. We obtained the expression of related genes in the normal tissues of patients with LUAD and cancer from the TCGA

database. KEGG and Gene Ontology (GO) analyses were conducted via the Database for Annotation, Visualization and Integrated Discovery (DAVID) web server (Huang da et al, 2009; Sherman et al, 2022). Clustered heatmaps for changing sites and regions were generated via the R package ComplexHeatmap (Gu et al, 2016), and clusters were generated via the Ward. D2 method (Chang et al, 2021). In addition, we used the R package pvclust to assess the uncertainty in hierarchical cluster analysis, with multiscale bootstrap with the number of bootstrap 1000.

## Gene selection and mRNA expression detection

Gene expression differences between primary tumor and normal samples in the TCGA-LUAD dataset were analyzed for all genes with shallow-changing CpG sites. Genes with significant differential expression were identified, and five genes that are likely involved in tumor-adjacent interactions were selected based on a review of relevant research.

RNA was extracted from stage 2 samples via TRIzol RNA isolation reagents following the instructions provided by Thermo Fisher Scientific. The extracted total RNA was quantified via Nanodrop (Thermo Fisher Scientific) and analyzed via 1% agarose gel electrophoresis (Sangon Biotech). cDNA was subsequently synthesized via the HiScript II 1st Strand cDNA Synthesis Kit (Vazyme Biotech Co., Ltd.) according to the standard protocol. Quantitative reverse transcription PCR (RT-qPCR) was performed in a 20 μl reaction mixture containing ChamQ Universal SYBR qPCR Master Mix (Vazyme Biotech Co., Ltd.) according to the following protocol: 60 s at 95 °C, followed by 40 cycles of 95 °C for 10 s and 60 °C for 30 s. The expression levels of all candidate genes were measured in triplicate, calculated via the cycle threshold (2-ΔCT) method, and normalized to the internal reference gene *ACTB*. A list of all the primers used in this study is provided in Table EV2.

## Hematoxylin–eosin staining and immunohistochemistry

LUAD tissue was fixed in 10% neutral buffered formalin, dehydrated through a graded ethanol series, and embedded in paraffin. Sections (6 μm) were cut, mounted, and dried at 60 °C for 1 h. Slides were deparaffinized in xylene, rehydrated through a descending ethanol gradient (100%, 95%, 85%, 75%), and rinsed in distilled water. Sections were stained with Harris hematoxylin for 5 min, rinsed, differentiated in 1% hydrochloric acid-ethanol for 3–5 s, and blued in 0.1% ammonia water or saturated lithium carbonate for 1 min. Eosin Y (1%) was applied for 2 min, followed by rapid dehydration in an ascending ethanol gradient (75%, 85%, 95%, 100%), clearing in xylene, and mounting with neutral balsam. Slides were scanned at 20× magnification using a digital pathology slide scanner.

Paraffin-embedded LUAD tissue sections (6 μm), prepared as described above, were deparaffinized in xylene (three changes, 5 min each) and rehydrated through a descending ethanol gradient (100%, 95%, 85%, 75%, 1 min each) followed by rinsing in distilled water. Antigen retrieval was performed by immersing sections in citrate buffer (pH 6.0) and heating in a microwave at high power for 10 min, followed by cooling to room temperature. Endogenous peroxidase activity was quenched with 0.3% hydrogen peroxide in methanol for 10 min, followed by three 5-min washes in

phosphate-buffered saline (PBS, pH 7.4). Sections were blocked with 5% goat serum in PBS at room temperature for 30 min to reduce nonspecific binding. The sections were then incubated with WNT7B primary antibody (Affinity Biosciences, DF9042, 1:100 dilution) overnight at 4 °C. After three 5-minute PBS washes, sections were incubated with HRP-conjugated secondary antibody at room temperature for 40 min. Following three additional PBS washes, 3,3'-diaminobenzidine (DAB) substrate was applied until a brown color developed (~1–3 min), and the reaction was stopped by rinsing in distilled water. Sections were dehydrated through an ascending ethanol gradient (75%, 85%, 95%, 100%, 1 min each), cleared in xylene (three changes, 2 min each), and mounted with neutral balsam. Stained slides were scanned at 20× magnification using a fully automated digital pathology scanner (LG-S80, Wuhan Servicebio Technology Co., Ltd.) to generate high-resolution digital images.

HE and H-DAB scan files were read and analyzed by QuPath (Bankhead et al, 2017) (v0.5.1). HE results were submitted to two experienced pathologists for independent tumor boundary delineation and exclusion of divergent locations. For H-DAB image analysis, stain vector estimation was performed on all objects in each sample before cell detection, and the cell DAB OD mean was used as a single threshold of 0.15 to delineate positive WNT7B cells. The distribution of cell DAB OD mean and the proportion of positive WNT7B cells in each ROI were statistically analyzed.

## Bisulfite amplicon sequencing (BSAS)

Genomic DNA was extracted from the tumor and adjacent tissues via the FastPure FFPE DNA Isolation Kit (Vazyme Biotech Co., Ltd.). Following extraction, the DNA was treated with bisulfite via the EZ DNA Methylation-Gold Kit (Zymo Research) according to the manufacturer's protocol. The bisulfite-converted DNA was then amplified via a two-stage PCR method with the TaKaRa Ex Taq Hot Start Version. The resulting PCR products were purified via the New Gel Mini Purification Kit (Zoman Biotech Co., Ltd.) and sequenced on the Illumina NovaSeq 6000 platform (GENE-SEEQ). The specific primers used for BSAS are detailed in Table EV2.

## Analysis of methylation trends, survival analysis, and model construction

Following the previously described RRBS analysis parameters, we conducted stringent quality control on the BSAS sequencing data and extracted the DNA methylation levels of CpG sites in the amplified regions (with over 1000× coverage for each target site in each sample). We calculated the relative difference in each CpG site between the tumor and adjacent tissues in each sample: the methylation level of the tumor sample was subtracted from the adjacent tissue value and divided by the tumor methylation level, resulting in a value between −100% and 100%.

To categorize differences, we established thresholds from 3 to 15% for defining high (steep change) and low (shallow change) differences. Patients were classified into high- or low-risk groups based on these thresholds. Kaplan–Meier survival curves were generated via the "survival" package to visualize cumulative survival times and the number of patients at risk at specified time points. All statistical analyses were performed within the R statistical environment (R version 4.4.1).

The Kaplan–Meier estimator, a nonparametric statistic, with log-rank test (Mantel–Cox) was used to calculate the cumulative survival time and compare the differences in OS between the two groups.

A combination of sites was used for further grouping: among five sites with consistent prognostic relevance, samples were classified into the shallow (high-risk) group if at least one, two, or three sites exhibited shallow changes, with separate analyses of the grouping effects.

## Single-cell transcriptomic analysis

We obtained a publicly available dataset GSE131907 (Data ref: Ahn and Lee, 2020; Kim et al, 2020) from the Gene Expression Omnibus (GEO) database, including tumor and paired-adjacent normal tissues from ten LUAD patients. Data processing and analysis were performed using Seurat (v5.1.0) in R, where low-quality genes (detected in <0.1% of cells) were filtered out, followed by log-normalization to TPM-like values and log2(TPM + 1) transformation. We identified 2,000 highly variable genes using the "FindVariableFeatures" function, performed principal component analysis (PCA) on these genes, and used the top 50 principal components for cell clustering via the "FindClusters" algorithm. Cell populations were annotated based on differentially expressed genes (DEGs) identified using "FindAllMarkers" (min.pct = 0.25, log2FC ≥ 0.25, adjusted $P$ value < 0.05) and canonical marker genes from literature, with results visualized through uniform manifold approximation and projection (UMAP). In addition, we used Seurat to compute pseudo-bulk expression profiles for each patient sample and iteratively re-calculated them after excluding specific cell types. Differential expression of target genes between tumor and adjacent normal tissues was analyzed using DESeq2. We further generated pseudo-bulk expression profiles for each annotated cell type, and performed sample clustering based on both the global gene expression profiles and the expression profiles of genes located in shallow or mixed methylation change regions identified in the first stage of analysis.

# Data availability

Deidentified data and related documents of this cohort study may be shared upon request with academic research colleagues. Each request will be evaluated and decided upon by the study board members. Please contact the corresponding author for further information.

The source data of this paper are collected in the following database record: biostudies:S-SCDT-10_1038-S44319-025-00612-4.

# Peer review information

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

## Acknowledgements

This work was supported by the National Key R&D Program of China (2021YFC2701603), the Science and Technology Development Fund, Macao S.A.R (FDCT) (0102/2024/AFJ), Fundamental Research Funds for the Central Universities (021414380507), the National High-Level Hospital Clinical Research Funding (2022-PUMCH-B-011, 2022-PUMCH-A-188), the Chinese Society of Clinical Oncology fund (Y-MSDPU2021-0190, Y-MSD2020-0270), the Beijing Health Promotion Association (BJHPA-FW-XHKT- 2020040400344), and the Ministry of Science and Technology of the People's Republic of China, Special Data Service for Oncology, and the National Population and Health Scientific Data Sharing Platform (NCMI-ABD02-201809, NCMI-YF02N-201906). We are grateful to the High-Performance Computing Center of Nanjing University for conducting the numerical calculations in this study on its blade cluster system. Additionally, we acknowledge all the members of Dr. Chen Jian-Qun's laboratory for their valuable comments and discussions. We also thank Cong Zhang, Yali Qin, Yang Song, Yuan Xu, Zhibo Zheng, Zhongxing Bing, Xinyu Liu, Xiaoqing Yu, Haochen Li, and Ruirui Li for their significant contributions.

## Author contributions

**Yifan Wu**: Data curation; Formal analysis; Validation; Investigation; Visualization; Writing—original draft. **Yadong Wang**: Resources; Data curation; Visualization; Writing—review and editing. **Yao Tang**: Validation; Investigation; Writing—review and editing. **Jianchao Xue**: Resources; Data curation; Writing—review and editing. **Zichen Jiao**: Investigation. **Bowen Li**: Data curation. **Sainan Wang**: Validation. **Zhicheng Huang**: Resources. **Xiaoyi Zheng**: Investigation; Visualization. **ChenZheng Guan**: Investigation. **Daoyun Wang**: Resources. **Ji Li**: Resources. **Lan Song**: Data curation. **Ka Luk Fung**: Conceptualization. **Heqing Xu**: Conceptualization. **Shanqing Li**: Conceptualization. **Liucun Zhu**: Conceptualization. **Jian-Qun Chen**: Conceptualization; Writing—review and editing. **David J Kerr**: Conceptualization; Writing—review and editing. **Naixin Liang**: Conceptualization; Project administration; Writing—review and editing. **Qiang Wang**: Supervision; Methodology; Writing—review and editing. **Qihan Chen**: Supervision; Project administration; Writing—review and editing.

Source data underlying figure panels in this paper may have individual authorship assigned. Where available, figure panel/source data authorship is listed in the following database record: biostudies:S-SCDT-10_1038-S44319-025-00612-4.

## Disclosure and competing interests statement

The authors declare no competing interests.

# Expanded View Figures

**Figure EV1.   Schematic diagrams illustrating the statistical methods employed.**

(A) Illustrations of spatial autocorrelation, featuring three typical distribution examples. (B) Illustration of the Kendall rank correlation coefficient. All points within the gray area are concordant, while those in the white area are discordant to the reference point. This example contains 395 concordant point pairs and 40 discordant pairs, yielding a Kendall rank correlation coefficient of 0.816. (C) The significantly different CpG site at chr1:1,138,200 is associated with no other significant sites within a 30 nt range. (D) Boxplots illustrate the methylation levels of the site chr1:1,138,200 across different locations, while matrices display the significantly associated locations for each CpG site. (E) The significantly different CpG site at chr2:42,905,798 is associated with three other significant sites within a 30 nt range. (F) Boxplots illustrate the methylation levels of these four sites across different locations, while matrices display the significantly associated locations for each CpG site. The methylation patterns of these sites are analogous, with chr2:42,905,798, chr2:42,905,801, and chr2:42,905,812 exhibiting comparable Moran's $I$ values. The matrices of these three surrounding sites are respectively compared with the calculated cosine similarity of chr2:42,905,798, and the results are shown on the blue lines. Boxplots present with the median as the center, box bounds as the 25th and 75th percentiles, and whiskers extending to the minimum and maximum values within 1.5×IQR ($n = 12$). Outliers beyond 1.5×IQR are plotted as individual points.

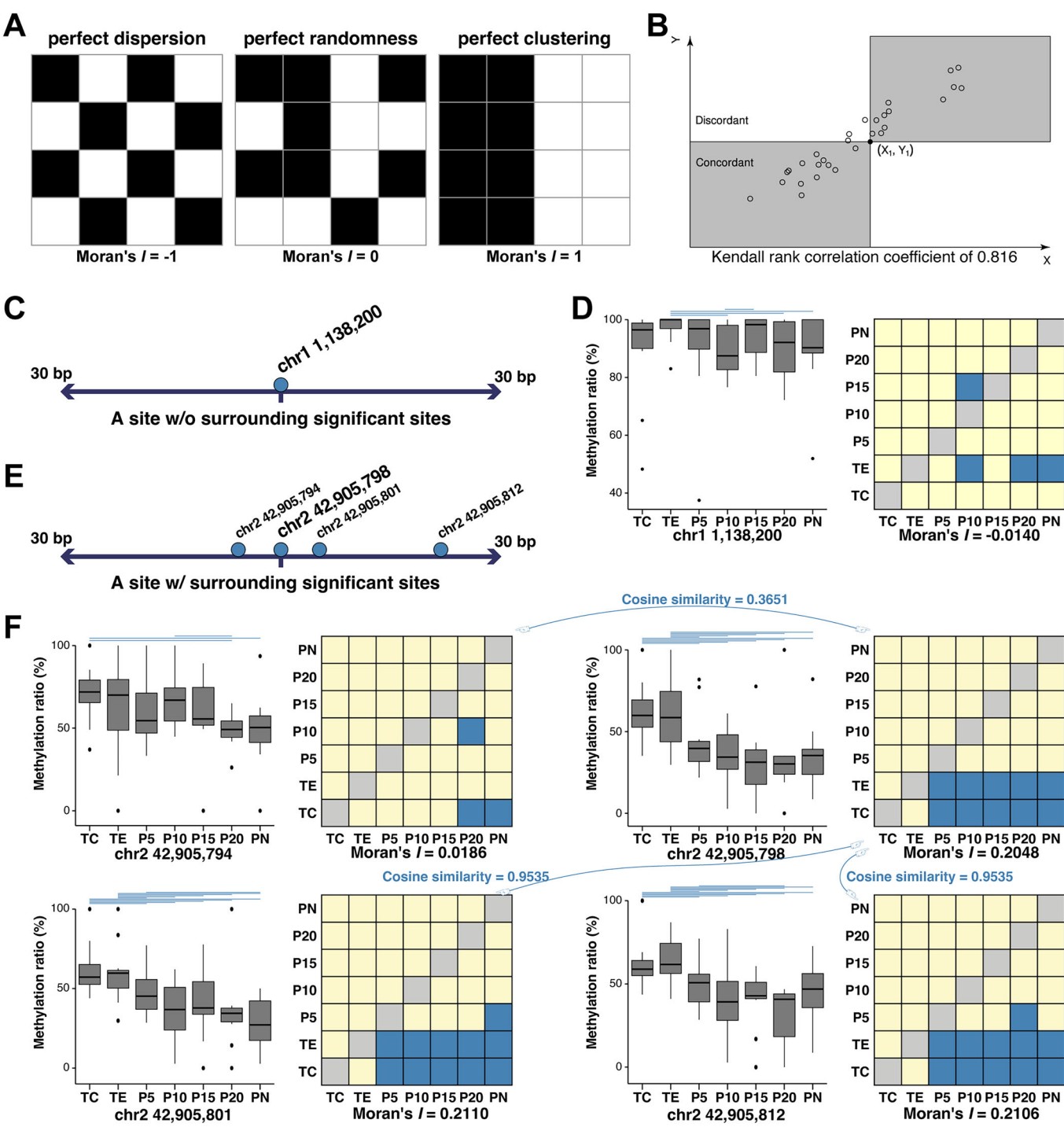

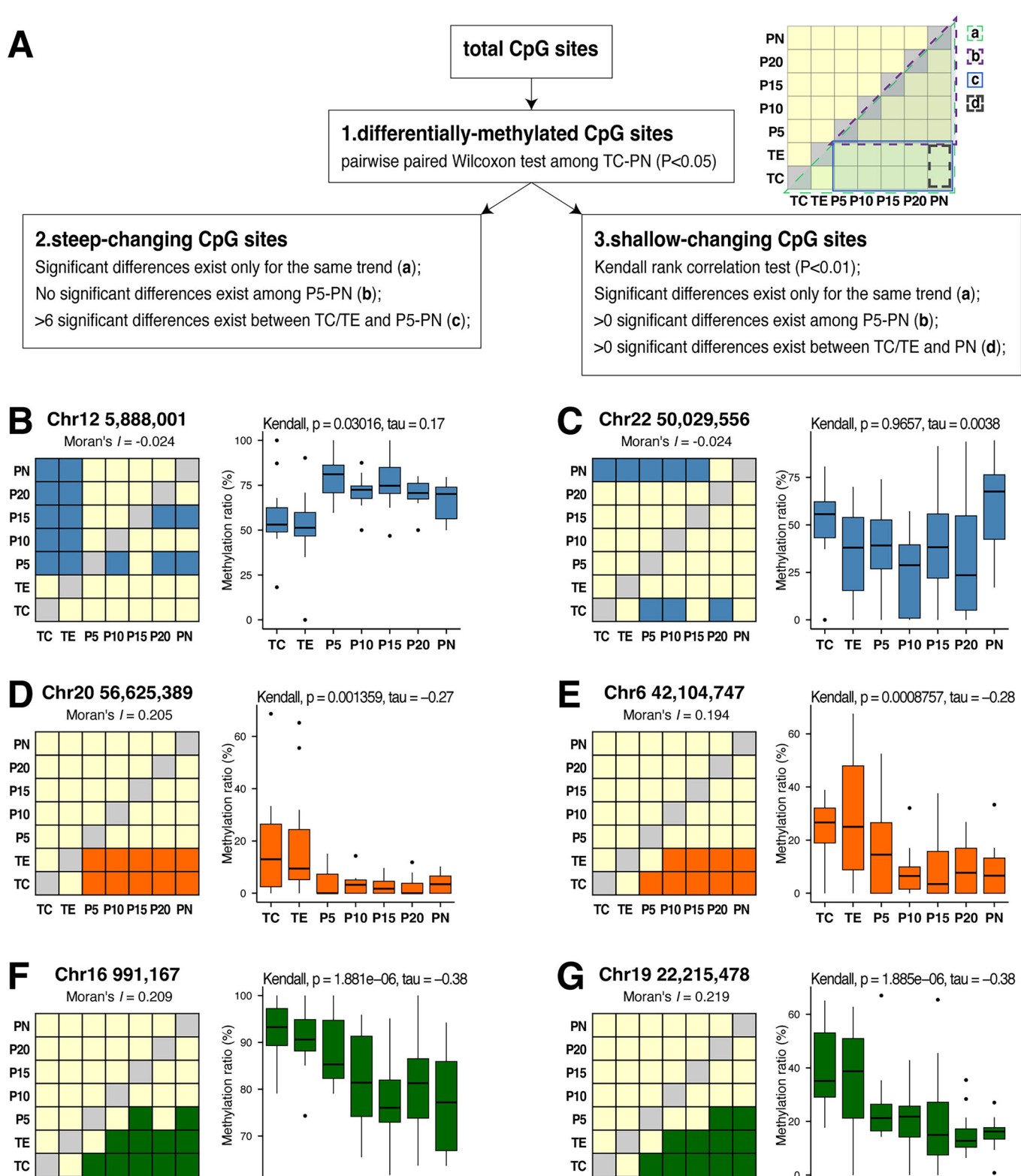

**Figure EV2. Classification process for methylation change trends and site examples.**

(A) The screening process and specific criteria for identifying steep- and shallow-changing sites are outlined. Additional details and examples of criteria for judgment can be found in Appendix Methods. (B, C) Two differentially methylated CpG sites were excluded for failing to meet the specified criteria. (D, E) Examples of two steep-changing CpG sites are presented. (F, G) Examples of two shallow-changing CpG sites are presented. Boxplots present with the median as the center, box bounds as the 25th and 75th percentiles, and whiskers extending to the minimum and maximum values within 1.5×IQR ($n = 12$). Outliers beyond 1.5×IQR are plotted as individual points.

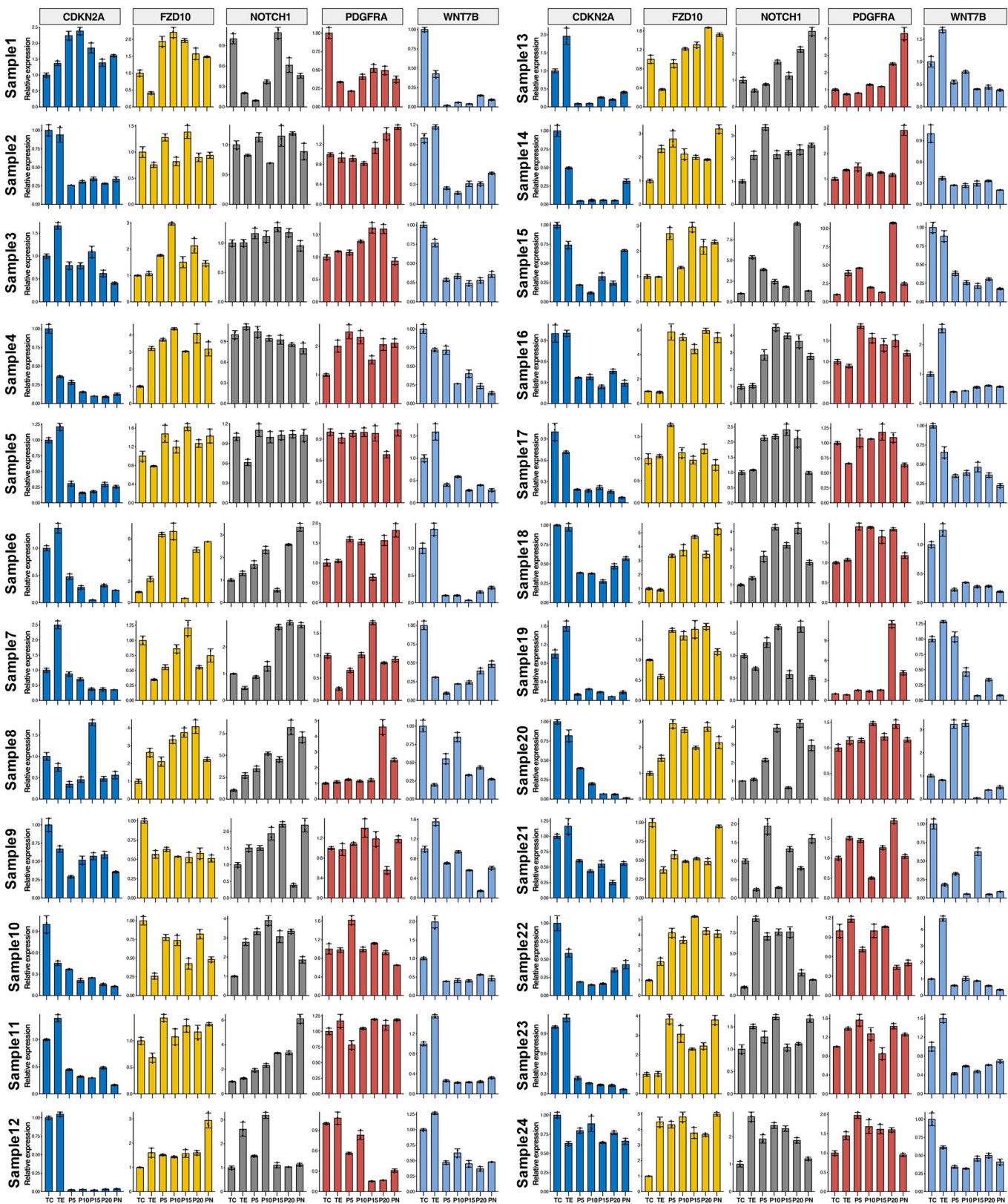

**Figure EV3.** Relative gene expression at different locations in each sample measured by RT-qPCR.

mRNA expression of five genes from 24 patients are presented. Relative expression levels were normalized to the reference gene *ACTB* and standardized to TC's relative expression for comparison, with error bars indicating the mean ± standard deviation (SD) from three replicates.

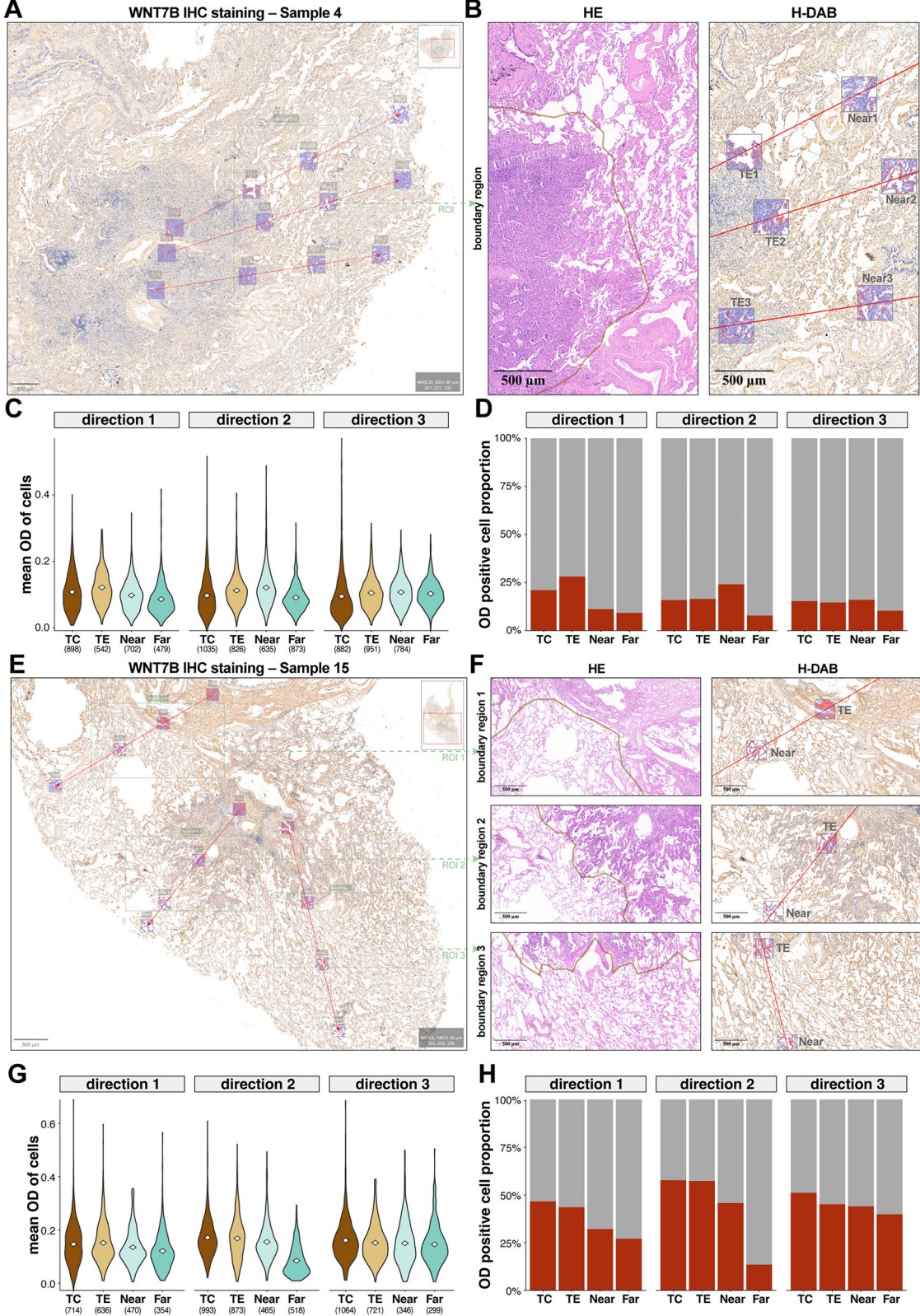

Figure EV4.   WNT7B protein expression profiles of Patient 4 and Patient 15.

(A) Representative IHC staining of WNT7B protein in LUAD Sample 4. FFPE tumor tissue section was stained using WNT7B antibody. Red lines marked three directions from the tumor core to adjacent tissues, and 300μm-squared regions of interest (ROIs) of the tumor core (TC), tumor edge (TE), and adjacent tissue at different distances (P5–Near and P5–Far) on each direction were selected as for optical density (OD) analysis. (B) A larger ROI covering three pairs of TE and P5-Near regions was selected for the local magnification display of the boundary region in Sample 4. HE and H-DAB staining images showed the visual boundary of the tumor and corresponding DAB signals in TE and P5-Near regions. (C) Violin plots showing the distribution of mean OD per cell for each region (TC, TE, Near, and Far) in three directions of Sample 4 and cell numbers of ROIs were shown below. (D) Proportion of OD-positive cells in each region and direction, based on predefined OD thresholds described in the Methods. (E) Representative IHC staining of WNT7B protein in LUAD Sample 15. FFPE tumor tissue section was stained using WNT7B antibody. Red lines marked three directions from the tumor core to adjacent tissues, and 300μm-squared regions of interest (ROIs) of the tumor core (TC), tumor edge (TE), and adjacent tissue at different distances (P5–Near and P5–Far) on each direction were selected as for optical density (OD) analysis. (F) Three larger ROIs covering the corresponding TE and P5-Near regions were selected for local magnification display of the boundary region in Sample 15. HE and H-DAB staining images showed the visual boundary of the tumor and corresponding DAB signals in TE and P5-Near regions. (G) Violin plots showing the distribution of mean OD per cell for each region (TC, TE, Near, and Far) in three directions of Sample 15 and cell numbers of ROIs were shown below. (H) Proportion of OD-positive cells in each region and direction in Sample 15.

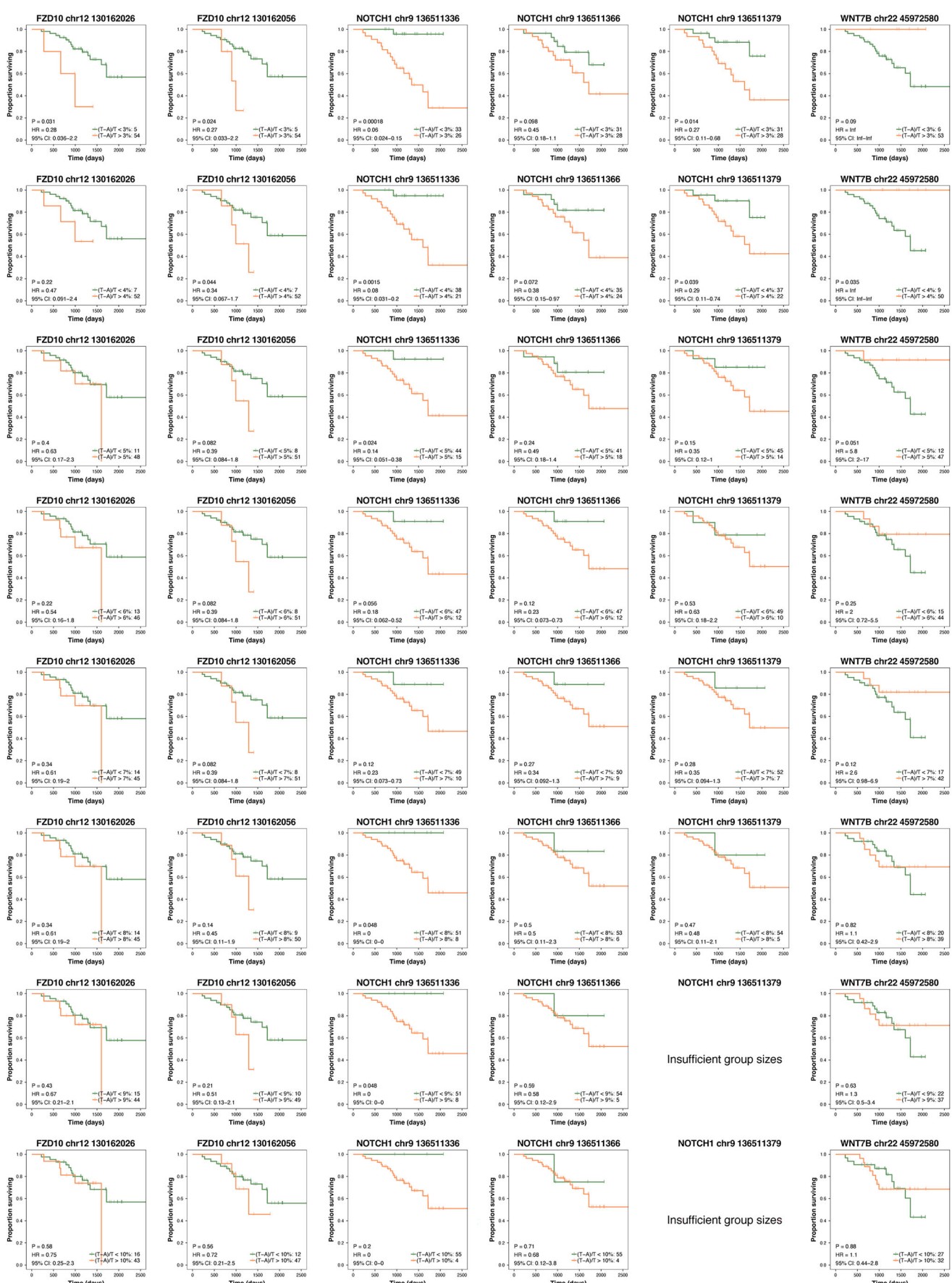

◄ **Figure EV5. Kaplan–Meier survival curves illustrating the relationship between shallow DNA methylation changes at individual CpG sites and overall survival, using 3% to 10% methylation difference thresholds.**

*P* values of K–M survival analysis were calculated using chi-squared distribution.

