## [Peer Review File · EMBO Reports]

Early-stage lung adenocarcinoma affects DNA methylation and gene expression in adjacent tissues

Yifan Wu, Yadong Wang, Yao Tang, Jianchao Xue, Zichen Jiao, Bowen Li, Sainan Wang, Zhicheng Huang, Xiaoyi Zheng, ChenZheng Guan, Daoyun Wang, Ji Li, Lan Song, Ka Luk Fung, Heqing Xu, Shanqing Li, Liucun Zhu, Jian-Qun Chen, David Kerr, Naixin Liang, Qiang Wang, and Qihan Chen

Corresponding author(s): Qihan Chen (lyonchen@um.edu.mo), Qiang Wang (wangq@nju.edu.cn), Naixin Liang (liangnaixin@pumch.cn)

Review Timeline:

Submission Date:	20th Dec 24
Editorial Decision:	13th Mar 25
Revision Received:	12th Jun 25
Editorial Decision:	8th Aug 25
Revision Received:	13th Aug 25
Accepted:	15th Sep 25

Editor: Bernd Pulverer / Martina Rembold

Transaction Report:

Dear Dr. Chen

Thank you for the submission of your manuscript to EMBO reports, and also for your patience in awaiting this delayed decision. We have now received two expert referee reports, pasted below.

As you will see, the referees acknowledge that the findings are potentially interesting if the main novel claim, that the 'shallow' gradient DNA methylation signature serves as a prognostic for high risk LUAD.

Referee 1 notes that the data is descriptive in the absence of any mechanistic insight into the TME spreading of the tumour DNA-methylation signature, which has been reported in outline before. However, the referee highlight the 'promising application for clinical prognosis'. Referee raises three key points:

- 1) the 3% threshold for distinction between 'steep'- and shallow'-gradient is unconvincing. The referee recommends to increase the threshold and expand to more CpG sites and to identify those most relevant as biomarkers, expanding from fig6B.
- 2) confirm the claim that gene expression serves as a proxy for DNA methylation, since this is much more amenable as a clinical biomarker assay by RT-PCR. Add control genes and expand RT-PCR genome wide.
- 3) provide a more detailed analysis of the DNA methylation changes underlying the steep vs. shallow signature, based on the existing dataset.

Referee 2 notes that the analysis is detailed, systematic and elegant. The referee requests more experimental detail for fig 2E, 2F and justification for the bins chosen. The referee, importantly, requests an unbiased analysis to support the claim that tissue in the vicinity of the tumour cannot be considered 'strictly normal'. Like referee 1, an expansion of the target genes is recommended.

Finally, a broadening of the analysis to existing sc data in NSCLC to report on the target genes is recommended, with a view of framing mechanistic hypotheses.

I would thus like to invite you to revise your manuscript with the understanding that the referee concerns must be fully addressed and their suggestions taken on board, by textual or, where appropriate, experimental revision.

Please address all referee concerns in a complete point-by-point response. Acceptance of the manuscript will depend on a positive outcome of a second round of review and editorial evaluation. It is EMBO reports policy to allow a single round of major revision only and acceptance or rejection of the manuscript will therefore depend on the completeness of your responses included in the next, final version of the manuscript.

We realize that it is difficult to revise to a specific deadline. In the interest of protecting the conceptual advance provided by the work, we recommend a revision within 3 months (that is by 13th Jun 2025). Please discuss the revision progress ahead of this time with me by e-mail or videocall, in particular if you require more time to complete the revisions or if certain recommended experiments turn out to be a challenge. We are generally happy to extend the revision time as required, assuming the conceptual advance is not compromised.

- 1) A data availability section providing access to data deposited in public databases is missing. If you have not deposited any data, please add a sentence to the data availability section that explains that.
- 2) Your manuscript contains statistics and error bars based on $n=2$. Please use scatter blots in these cases. No statistics should be calculated if $n=2$.

3) We replaced Supplementary Information with Expanded View (EV) Figures and Tables that are collapsible/expandable online. A maximum of 5 EV Figures can be typeset. EV Figures should be cited as 'Figure EV1, Figure EV2' etc... in the text and their respective legends should be included in the main text after the legends of regular figures.

5) a complete author checklist, which you can download from our author guidelines <<https://www.embopress.org/page/journal/14693178/authorguide>>. Please insert information in the checklist that is also reflected in the manuscript. The completed author checklist will also be part of the RPF.

6) Please note that all corresponding authors are required to supply an ORCID ID for their name upon submission of a revised manuscript (<<https://orcid.org/>>). Please find instructions on how to link your ORCID ID to your account in our manuscript tracking system in our Author guidelines <<https://www.embopress.org/page/journal/14693178/authorguide#authorshipguidelines>>

Discussion of statistical methodology can be reported in the materials and methods section, but figure legends should contain a

basic description of n, P and the test applied.

12) All Materials and Methods need to be described in the main text using our 'Structured Methods' format, which is required for all research articles. According to this format, the Methods section includes a Reagents and Tools Table (listing key reagents, experimental models, software and relevant equipment and including their sources and relevant identifiers) followed by a Methods and Protocols section describing the methods using a step-by-step protocol format. The aim is to facilitate adoption of the methodologies across labs. More information on how to adhere to this format as well as a downloadable template (.docx) for the Reagents and Tools Table can be found in our author guidelines: <https://www.embopress.org/page/journal/14693178/authorguide#structuredmethods>.

An example of a Method paper with Structured Methods can be found here: <https://www.embopress.org/doi/full/10.1038/s44320-024-00037-6#sec-4>

I look forward to seeing a revised form of your manuscript when it is ready.

Yours sincerely,

~~~~~  
Bernd Pulverer, Ph.D.  
Chief Editor, EMBO Reports  
EMBO  
Meyerhofstrasse 1, D-69117 Heidelberg  
Tel: +4962218891501  
[bernd.pulverer@embo.org](mailto:bernd.pulverer@embo.org)  
~~~~~

Referee #1:

DNA methylation plays a critical role in cancer development and serves as a biomarker for cancerous signatures. Recently, the tumor microenvironment (TME) has also been found to exhibit cancer-associated signatures, potentially contributing to tumor homeostasis. A recent study demonstrated that the TME shares DNA methylation characteristics with tumor cells in lung adenocarcinoma (LUAD). Building on this prior work, Wu et al. aimed to compare the DNA methylation landscape between tumor cells and neighboring cells at increasing distances from the tumor boundary. To achieve this, they performed DNA methylation genomic mapping using Reduced Representation Bisulfite Sequencing (RRBS) on an initial cohort of 12 early-stage LUAD patients. This analysis revealed over 17,000 differentially methylated CpG between tumor cells and neighboring cells. These DNA methylation changes were classified into two categories: « steep » changes, where DNA methylation alterations occurred at the tumor-microenvironment boundary, and « shallow » changes, where DNA methylation gradually shifted over increasing physical distance from the tumor boundary. The authors validated these findings in a larger cohort of 24 patients by examining gene expression changes associated with DNA methylation alterations observed in the initial cohort and found that these gene expressions follow the same shallow-changing trends. Finally, using a four-year follow-up 59 patients-cohort, they suggested that specific "shallow" DNA methylation signatures could serve as prognostic biomarkers, distinguishing high-risk from low-risk patients. While this study is primarily descriptive and does not explore the molecular mechanisms underlying these DNA methylation changes, it highlights a promising application for clinical prognosis. However, several aspects of the study have to

be improved prior to publication at EMBO reports:

Major Points:

1. The threshold of 3% used to distinguish between steep and shallow DNA methylation change in a unique CpG is not appropriate to clinically discriminate between low and high-risks patients. This threshold might reflect small physiological variations, such as fluctuations in cell composition or technical biases (e.g., bisulfite conversion rates, which can vary by up to 2%). To improve the prognostic strategy, the authors should increase this threshold / expand the number of CpG sites included in the analysis. Global DNA methylation analysis (such as RRBS) may be beneficial for identifying the most relevant CpG sites or regions to use as biomarkers in FFPE sample preparation, as represented in figure 6B. Furthermore, as the author showed that DNA methylation correlates with gene expression change, can the same prognosis strategy be applied with RT-qPCR instead of DNA methylation? (This approach would even further decrease the prognosis costs). Given that this is the key contribution of the study to the literature, the authors should strengthen this point.
2. The authors should provide a more detailed analysis of the DNA methylation changes that characterize steep and shallow methylation patterns. While some information is included in the supplementary tables, the genomic context of these changes is not thoroughly explored. Specifically, comparison of the genomic distribution of CpG methylation changes, overlap with CpG islands or transcription factor binding sites may be beneficial to generate hypotheses regarding the mechanisms underlying shallow DNA methylation changes.
3. The authors have shown that shallow DNA methylation change trend correlate with the trend of gene expression changes for several target genes. This observation should be validated genome wide by performing RNA-seq. Additionally, the authors should include additional housekeeping genes for RT-qPCR normalization.
4. The paper suffers from a lack of clarity in both text and figures, which may hinder reader comprehension. The author should overall improve the quality of the paper. Specifically, Figure legends are sometimes missing, particularly for heatmaps. Several supplementary figure panels are entirely absent, making it difficult to assess supporting data. Effort should be made to clearly visualize the data (supp fig 10). The text is sometimes unclear, or overstated for example:
Line 92: LUAD instead of "LAUD"
Line 142: "We analyzed a total of 613,724 CpG sites, accounting for approximately 40% of all CpG sites, and identified at least two significant difference sites between regions." Do the author mean that they identified 613 724 CpG differentially methylated? These count for 40% of analyzed CpG not all the CpGs of the genome.
Line 197: "DNA methylation sites typically do not operate independently; rather, they function in concert through the formation of CpG islands, facilitating the regulation of associated genes." This sentence should be reformulated for clarity.

Referee #2:

In this study, the authors examined the impact of tumors on adjacent tissues from the perspective of CpG gradient angles. Using a discovery cohort of 12 patients, they identified approximately 17,000 sites exhibiting a shallow change pattern and selected five genes with shallow-changing regions to investigate their gene expression and impact on patient overall survival (OS) in a larger validation cohort. The authors conducted a systematic and detailed analysis of CpG sites, carefully selecting each cutoff to categorize CpG sites, regions, and patients into distinct groups. Additionally, leveraging Moran's I index to assess CpG change patterns is a smart and elegant approach. I have the following questions that need to be addressed:

1. In lines 149-156, 11 bins were selected to analyze the Moran's I distribution patterns. What would be the impact of using a different number of bins? Additionally, more methodological details should be provided to explain how Figures 2E and 2F were generated. For instance, how was the similarity in methylation status among different CpG sites, as shown in Supplementary Figure 2D, calculated?
2. In lines 190-195, since the clustering was performed using only shallow-changing sites, it is expected that TE would cluster with P5/P10. Likewise, if only steep-changing sites were used, adjacent tissues from P5 to PN would likely cluster together. Therefore, to support the conclusion in lines 193-194 that "adjacent tissues within 10 mm of the tumor boundary may no longer be considered strictly normal," some form of unbiased quantification or clustering analysis should be applied.
3. The authors have not provided sufficient details on how the five target genes were selected based on steep/shallow-changing trends. It is suggested to establish a gene selection pipeline that includes all genes meeting the defined criteria and assess their functional relevance. Additionally, it would be valuable to explore single-cell data from NSCLC to determine which cell types express these target genes. For instance, PDGFRA is a specific marker for fibroblasts, so investigating whether these genes are associated with the tumor microenvironment would be of interest and provide insights into the mechanisms driving CpG site changes from tumor to adjacent regions..
4. As mentioned in line 102, the authors should provide an example of an HE image in the supplementary figures.
5. A color bar should be included in Figures 2A-2C for clarity.
6. It is recommended to remove the bottom heatmap panel in Figure 5A, as it presents redundant information and may complicate interpretation.

Point-by-point Response to Reviewers

Referee #1:

DNA methylation plays a critical role in cancer development and serves as a biomarker for cancerous signatures. Recently, the tumor microenvironment (TME) has also been found to exhibit cancer-associated signatures, potentially contributing to tumor homeostasis. A recent study demonstrated that the TME shares DNA methylation characteristics with tumor cells in lung adenocarcinoma (LUAD). Building on this prior work, Wu et al. aimed to compare the DNA methylation landscape between tumor cells and neighboring cells at increasing distances from the tumor boundary. To achieve this, they performed DNA methylation genomic mapping using Reduced Representation Bisulfite Sequencing (RRBS) on an initial cohort of 12 early-stage LUAD patients. This analysis revealed over 17,000 differentially methylated CpG between tumor cells and neighboring cells. These DNA methylation changes were classified into two categories: « steep » changes, where DNA methylation alterations occurred at the tumor-microenvironment boundary, and « shallow » changes, where DNA methylation gradually shifted over increasing physical distance from the tumor boundary. The authors validated these findings in a larger cohort of 24 patients by examining gene expression changes associated with DNA methylation alterations observed in the initial cohort and found that these gene expressions follow the same shallow-changing trends. Finally, using a four-year follow-up 59 patients-cohort, they suggested that specific "shallow" DNA methylation signatures could serve as prognostic biomarkers, distinguishing high-risk from low-risk patients. While this study is primarily descriptive and does not explore the molecular mechanisms underlying these DNA methylation changes, it highlights a promising application for clinical prognosis. However, several aspects of the study have to be improved prior to publication at EMBO reports:

We greatly appreciate the reviewer's thoughtful and positive comments and feedback. We believe that the suggestions are constructive and helpful for enhancing the quality of the manuscript. Please see below our point-to-point responses to the comments. For

your convenience, we have included the corresponding panels immediately below our responses. In the revised manuscript, the revised content has been marked with a red color, and all of the new panels have been incorporated.

Major Points:

1. The threshold of 3% used to distinguish between steep and shallow DNA methylation change in a unique CpG is not appropriate to clinically discriminate between low and high-risks patients. This threshold might reflect small physiological variations, such as fluctuations in cell composition or technical biases (e.g., bisulfite conversion rates, which can vary by up to 2%). To improve the prognostic strategy, the authors should increase this threshold / expand the number of CpG sites included in the analysis. Global DNA methylation analysis (such as RRBS) may be beneficial for identifying the most relevant CpG sites or regions to use as biomarkers in FFPE sample preparation, as represented in figure 6B. Furthermore, as the author showed that DNA methylation correlates with gene expression change, can the same prognosis strategy be applied with RT-qPCR instead of DNA methylation? (This approach would even further decrease the prognosis costs). Given that this is the key contribution of the study to the literature, the authors should strengthen this point.

Response:

Thank you for your thoughtful comments and valuable suggestions. Your feedback touches on several important aspects, and we would like to address each point in turn.

First, regarding the use of a threshold to distinguish between steep and shallow DNA methylation changes in the third stage of analysis: Following our exploratory findings in the first and second stages, we hypothesized that alterations in adjacent tissues could reflect underlying tumor biology and potentially influence patient prognosis. To ensure clinical applicability, we opted to assess only two regions—tumor and its immediately adjacent tissue—focusing on specific CpG sites. The methylation difference between these two sites was used to infer whether the tumor had extended influence into

surrounding tissues. While this approach differs from the full seven-region analysis used earlier, it serves as a clinically feasible method to evaluate the prognostic relevance of shallow methylation changes.

We appreciate your concern that a 3% difference might fall within the range of technical variability, such as fluctuations in bisulfite conversion efficiency. Initially, we tested thresholds of 3%, 5%, and 10%, and presented results for 3% based on preliminary prognostic separation. However, we acknowledge that we did not provide sufficient justification for this choice. In the revised manuscript, we have systematically evaluated thresholds ranging from 3% to 15%, at 1% intervals (in lines 346-359). For each threshold, we assessed the prognostic performance of six CpG sites individually and in combination using Kaplan-Meier survival analysis (Figure 6C). Notably, five sites on *FZD10* and *NOTCH1* exhibited high HRs at lower thresholds. Their HRs decreased as the threshold rose from 3% to 7%, with ranges as follows: chr12:130,162,026 (HR: 1.6–3.6), chr12:130,162,056 (HR: 2.6–3.7), chr9:136,511,336 (HR: 4.3–17), chr9:136,511,366 (HR: 2.1–4.3), and chr9:136,511,379 (HR: 1.6–3.7), and the site on *WNT7B* (chr22:45,972,580) displayed the opposite trend: low-difference groups were associated with better prognosis, with HRs decreasing from 0 to 0.49 as the threshold increased from 3% to 7%.

Figure 6C. Risk stratification performance with different difference ratios as thresholds at each site.

While raising the threshold may reduce the impact of technical noise, it could also misclassify some biologically relevant “steep” changes as “shallow,” thereby reducing prognostic power. To strike a balance, and in response to your recommendation, we have updated the main analysis to use a 5% threshold in the revised manuscript to mitigate technical variation while preserving biological signal (in lines 360-366).

Second, regarding your suggestion to expand CpG site selection using genome-wide methylation profiling (e.g., RRBS) to build a more comprehensive predictive model: We completely agree. However, the aim of the current study was not to identify the optimal set of prognostic markers, but rather to explore a biological phenomenon—namely, shallow methylation changes in adjacent tissues—and to investigate their associations with gene expression, protein levels, and patient outcomes. Thus, the candidate genes assessed in the second and third stages were derived from this biological context. That said, we acknowledge the importance of systematic marker discovery. In our previous work, we have developed and validated robust methylation-based prognostic models for several cancer types using genome-wide datasets (Guo et al, 2018; Guo et al, 2019). In the future, we plan to integrate this approach into a larger, multi-center cohort study currently underway, which will allow for more rigorous selection and validation of CpG biomarkers based on high-throughput methylation data.

Third, on your recommendation to consider RT-qPCR as a cost-effective alternative to DNA methylation assays in clinical settings: We fully endorse this idea. Based on our second-stage results, mRNA expression levels of several target genes correlated well with methylation changes, suggesting that gene expression measurements from tumor and adjacent tissues could provide a rapid and clinically feasible proxy for assessing tumor behavior. However, retrospective validation of this strategy is constrained by sample limitations—specifically, the need for matched tumor and adjacent tissue, high-quality RNA, and longer than 4 years follow-up data of LUAD samples. Furthermore, while RT-qPCR offers a lower-cost approach, mRNA is inherently more labile and

sensitive to degradation or biological fluctuations than DNA methylation, potentially introducing noise. These perspectives have been incorporated into the revised manuscript under the discussion section (in lines 558-564), as we appreciate your emphasis on strengthening the clinical utility of our findings.

2. The authors should provide a more detailed analysis of the DNA methylation changes that characterize steep and shallow methylation patterns. While some information is included in the supplementary tables, the genomic context of these changes is not thoroughly explored. Specifically, comparison of the genomic distribution of CpG methylation changes, overlap with CpG islands or transcription factor binding sites may be beneficial to generate hypotheses regarding the mechanisms underlying shallow DNA methylation changes.

Response:

We thank the reviewer for this valuable suggestion. In our original submission, we did not provide sufficient analysis or discussion of the genomic context related to steep and shallow methylation patterns, and your guidance has greatly helped us enhance this aspect of the study. In the revised manuscript, we have incorporated a more detailed analysis of the distribution of steep and shallow methylation-changing CpG sites across various genomic elements, including gene bodies, promoter regions (defined as 2 kb upstream of the transcription start site), CpG islands, TF binding regions, and CTCF binding regions.

As shown in the figure above, compared with other CpG sites, steep/shallow changing sites are less likely to be located in promoter regions while both were relatively enriched in TF binding regions. We further stratified the changing sites by rise and decline trends, and found that both steep and shallow rise sites were less frequently located in CpG islands, while steep decline sites were more commonly located within these regions. In TF binding regions, gain-of-methylation events were more frequently observed, with 2.30% of steep rise and 1.97% of shallow rise sites located in these regions, several times higher than other CpG sites, and the proportion of decline sites was also higher

than that of other CpG sites. These findings suggest that such methylation alterations tend to occur in regions involved in transcriptional regulation, where increased methylation may interfere with transcription factor binding, leading to gene downregulation. We have included the related descriptions and discussions in the revised manuscript (in lines 182-197).

Appendix Figure S3. Genomic context distribution of CpG sites with steep and shallow methylation changes. (A) Comparison of CpG sites with steep ($n = 17,794$, orange) and shallow ($n = 17,498$, green) methylation changes against all other detected CpG sites ($n = 1,495,301$, grey) across five genomic features: gene bodies, promoters (2 kb upstream of TSS), CpG islands, transcription factor (TF) binding regions, and CTCF binding regions. (B) Feature distribution of steep rise and steep decline sites. (C) Feature distribution of shallow rise and shallow decline sites. Site counts and their corresponding proportions are labeled on each bar.

3. The authors have shown that shallow DNA methylation change trend correlate with the trend of gene expression changes for several target genes. This observation should be validated genome wide by performing RNA-seq. Additionally, the authors should include additional housekeeping genes for RT-qPCR normalization.

Response:

We appreciate your insightful comments regarding the correlation between shallow DNA methylation changes and gene expression, and we fully agree that genome-wide RNA-seq analysis would provide a broader and more systematic validation of this relationship. As noted in our response to Comment 1, the primary aim of our study was to characterize a spatially progressive (shallow change) methylation phenomenon and explore its potential biological implications. Accordingly, our mRNA validation (Stage 2) and prognostic evaluation (Stage 3) focused on a set of candidate genes identified based on the methylation patterns observed in Stage 1 to prove of concept rather than employing an unbiased genome-wide screening approach at the transcriptomic level.

To further explore whether the observed mRNA changes are also reflected at the protein level, we incorporated immunohistochemical (IHC) staining to assess protein-level changes in selected samples. Consistently, across multiple directions in several tumor samples, we did not observe steep changes in the expression of WNT7B protein at the tumor's visual boundary (in lines 303-324). This additional evidence helps to more comprehensively illustrate the potential influence of tumor tissues on their surrounding environment and reinforces the coherence of our overall findings.

We believe that integrating genome-wide RNA-seq analysis would offer a valuable and complementary perspective for further dissecting the consistency and divergence between DNA methylation and transcriptional regulation across tumor-adjacent gradients. We are actively considering such approaches for future investigations.

Regarding the selection of housekeeping genes for RT-qPCR normalization, we used *ACTB* as an internal control. This choice stemmed from a comparative analysis of

multiple housekeeping genes in the TCGA-LUAD dataset, where *ACTB* demonstrated stable expression across samples, whereas commonly used genes like *GAPDH* exhibited significant variability. We have clarified this matter in the revised manuscript text (in lines 274-276). We apologize for the previous lack of clarity and sincerely thank you for bringing this to our attention.

4. The paper suffers from a lack of clarity in both text and figures, which may hinder reader comprehension. The author should overall improve the quality of the paper. Specifically, Figure legends are sometimes missing, particularly for heatmaps. Several supplementary figure panels are entirely absent, making it difficult to assess supporting data. Effort should be made to clearly visualize the data (supp fig 10). The text is sometimes unclear, or overstated for example:

Line 92: LUAD instead of "LAUD"

Line 142: "We analyzed a total of 613,724 CpG sites, accounting for approximately 40% of all CpG sites, and identified at least two significant difference sites between regions." Do the author mean that they identified 613 724 CpG differentially methylated? These count for 40% of analyzed CpG not all the CpGs of the genome.

Line 197: "DNA methylation sites typically do not operate independently; rather, they function in concert through the formation of CpG islands, facilitating the regulation of associated genes." This sentence should be reformulated for clarity.

Response:

Thank you for your constructive feedback. We have carefully reviewed and revised the text to correct graphical errors and clarify ambiguous or imprecise statements, including the issues highlighted on lines 92, 142, and 197. In response to your comments on figure clarity, we have redrawn the relevant figures and added missing figure legends, particularly for heatmaps and supplementary figure panels that were previously incomplete due to layout errors. We also made adjustments to figure formatting and visualization to enhance readability and overall presentation quality. We hope these improvements have addressed the issues that may have hindered interpretation in the earlier version. If any further inconsistencies remain, we sincerely welcome additional suggestions and will revise accordingly.

Referee #2:

In this study, the authors examined the impact of tumors on adjacent tissues from the perspective of CpG gradient angles. Using a discovery cohort of 12 patients, they identified approximately 17,000 sites exhibiting a shallow change pattern and selected five genes with shallow-changing regions to investigate their gene expression and impact on patient overall survival (OS) in a larger validation cohort. The authors conducted a systematic and detailed analysis of CpG sites, carefully selecting each cutoff to categorize CpG sites, regions, and patients into distinct groups. Additionally, leveraging Moran's I index to assess CpG change patterns is a smart and elegant approach. I have the following questions that need to be addressed:

We greatly appreciate the reviewer's thoughtful and positive comments and feedback. We believe that the suggestions are constructive and helpful for enhancing the quality of the manuscript. Please see below our point-to-point responses to the comments. For your convenience, we have included the corresponding panels immediately below our responses. In the revised manuscript, the revised content has been marked with a red color, and all of the new panels have been incorporated.

1. In lines 149-156, 11 bins were selected to analyze the Moran's I distribution patterns. What would be the impact of using a different number of bins? Additionally, more methodological details should be provided to explain how Figures 2E and 2F were generated. For instance, how was the similarity in methylation status among different CpG sites, as shown in Supplementary Figure 2D, calculated?

Response:

We sincerely apologize for any confusion caused by the lack of detailed and precise methodological description in the original manuscript. In our initial analysis of Moran's I distribution, we divided the values into 11 equal-width bins starting from 0, using intervals of 0.025. The lowest and highest bins included all values ≤ -0.025 and > 0.20 , respectively.

In the revised version, following your suggestion, we tested alternative binning strategies to evaluate the robustness of the observed patterns (in lines 630-639). These included:

- Using finer (0.0125) and coarser (0.05) fixed-width intervals, resulting in 21 and 6 bins respectively;
- Quantile-based binning with 10 and 20 unequal-width bins based on deciles and vigintiles;
- Data-driven classification using Gaussian Mixture Models (GMM) with expectation-maximization (EM), and the Fisher-Jenks natural breaks optimization method based on minimizing within-group variance.

As shown in the figure below, across all binning methods, the trends remained consistent: CpG sites with higher Moran's I values—indicating stronger spatial autocorrelation—tended to have more neighboring differentially methylated sites within ± 30 nt, and higher similarity in differential patterns (measured as cosine similarity of binary significance vectors).

In addition, we have updated Methods of the revised manuscript to include detailed descriptions of these analytical procedures (in lines 640-649). Another illustrative example has also been added to Figure EV1 to clarify how local similarity was calculated, based on cosine similarity between binary vectors representing site-specific differential status across all pairwise comparisons.

Appendix Figure S2. Robustness of the association between Moran's I and spatial ordering of sites across different binning strategies. To evaluate the robustness of the observed correlation between Moran's I and spatial ordering of CpG sites, six binning strategies were applied: (A) fixed-width bins at 0.0125 intervals (21 bins), (B) fixed-width bins at 0.05 intervals (6 bins), (C) equal-sized quantile bins (10 bins), (D) quantile-based bins with finer resolution (20 bins), (E) bins generated by Gaussian mixture model (GMM) using the expectation-maximization (EM) algorithm (12 bins), and (F) bins defined by Fisher-Jenks natural breaks optimization (11 bins).

2. In lines 190-195, since the clustering was performed using only shallow-changing sites, it is expected that TE would cluster with P5/P10. Likewise, if only steep-changing sites were used, adjacent tissues from P5 to PN would likely cluster together. Therefore, to support the conclusion in lines 193-194 that "adjacent tissues within 10 mm of the tumor boundary may no longer be considered strictly normal," some form of unbiased quantification or clustering analysis should be applied.

Response:

Thank you very much for your valuable suggestion. In response, we performed a series of hierarchical clustering analyses across four different feature sets to provide a more comprehensive and unbiased assessment of the separation between tumor and adjacent tissues (in lines 430-445). These sets included: (1) all ~1.53 million CpG sites; (2) the filtered set of ~610,000 sites with at least two significant differences used in the Moran's I calculation; (3) the union of steep and shallow changing sites ($n = 35,292$); and (4) steep or shallow changing sites alone, as done previously. In addition, we conducted a resampling-based robustness test using "pvclust" (<https://github.com/shimo-lab/pvclust>) and performed bin-wise clustering based on Moran's I to further explore methylation trends at varying levels of spatial correlation.

First, hierarchical clustering using the complete set of ~1.53 million CpG sites (Appendix Figure S7A) revealed no clear tumor-normal separation. This suggests that the specific methylation trends we observed are confined to a relatively small or functionally critical subset of CpG sites. Second, using the filtered ~610,000 sites (Appendix Figure S7B), we observed that most tumor samples clustered distinctly; however, two TE samples grouped with adjacent tissues. This may reflect strong patient-specific methylation signatures in those tumors, rather than generalized tumor-specific patterns.

When steep and shallow changing sites were combined ($n = 35,292$; Appendix Figure S7C), tumor-normal separation became clearer, likely driven by the large effect sizes of steep changing sites. In this context, the subtle differences captured by shallow changing sites may have been masked—potentially explaining why such gradual

patterns are often overlooked in conventional analyses. In contrast, clustering based solely on shallow changing sites revealed that a subset of adjacent tissues (P5/P10) clustered together with certain tumor regions, consistent with our earlier observation.

Appendix Figure S7A. Clustering of samples using all ~1.53 million CpG sites did not reveal distinct tumor-normal separation.

Appendix Figure S7B. Clustering based on ~610,000 filtered CpGs (with ≥ 2 significant differences) showed clearer tumor grouping, though two TE samples clustered with adjacent tissues.

Appendix Figure S7C. Clustering using the combined set of steep and shallow changing CpG sites (n = 35,292).

To further support the robustness of our clustering conclusions, we applied the R package “pvclust” with multiscale bootstrap resampling to provide unbiased estimates of cluster stability (Appendix Figure S7D). Notably, P5 and P10 samples that clustered with tumors exhibited relatively high AU (Approximately Unbiased) p-values, supporting the

reliability of their assignments and helping mitigate potential overinterpretation based solely on classical hierarchical clustering.

Appendix Figure S7D. Cluster stability was assessed using pvclust with multiscale bootstrap resampling.

Finally, we explored whether similar trends could be observed by stratifying CpG sites based on their spatial autocorrelation (Moran's I) values. Using fixed-width binning (interval = 0.025, 11 bins), we observed a clear progression in clustering structure with increasing Moran's I : lower bins yielded disordered patterns, while bins 8, 9, and 11 showed robust tumor-normal separation (the figure below). Interestingly, in bin 10, one P5 sample grouped with tumor samples—a sample that also clustered with tumors in the shallow changing site analysis.

Clustering using CpG sites in each bin from Moran's I distribution, with a fixed-width interval 0.025.

To further refine the analysis, we used finer binning (interval = 0.0125, 21 bins; the figure below). From bin 14 onward, clear tumor-normal clustering emerged. In bins 19 and 20, one and two P5 samples, respectively, clustered with tumor tissues. These results suggest that even within subsets of CpG sites defined solely by high spatial autocorrelation (rather than by predefined steep/shallow patterns), some adjacent tissues exhibit tumor-like methylation signatures—highlighting that such observations are not artifacts of prior classification criteria but reflect biologically relevant transitions.

Clustering using CpG sites in each bin from Moran's I distribution, with a finer fixed-width interval of 0.0125.

3. The authors have not provided sufficient details on how the five target genes were selected based on steep/shallow-changing trends. It is suggested to establish a gene selection pipeline that includes all genes meeting the defined criteria and assess their functional relevance. Additionally, it would be valuable to explore single-cell data from NSCLC to determine which cell types express these target genes. For instance, PDGFRA is a specific marker for fibroblasts, so investigating whether these genes are associated with the tumor microenvironment would be of interest and provide insights into the mechanisms driving CpG site changes from tumor to adjacent regions.

Response:

Thank you for your insightful comments. We have now added a detailed explanation in Methods to clarify the rationale for selecting the five target genes (in lines 681-684).

Specifically, we first prioritized genes located within genomic regions enriched for shallow-changing CpG sites (either by local regions or site-level annotations). These candidate genes were then further filtered based on differential expression between tumor and normal tissues in the TCGA-LUAD dataset and evaluated for prognostic relevance via survival analysis. Finally, we cross-referenced relevant literature to select five representative genes—*CDKN2A*, *FZD10*, *NOTCH1*, *PDGFRA*, and *WNT7B*—which are potentially involved in tumor-adjacent interactions and are supported by prior findings.

We fully agree that integrating single-cell data adds important biological context. Ideally, we sought datasets that matched our specific spatial sampling strategy—including distal normal tissue, tumor-adjacent regions, and tumor core regions—to accurately investigate how tumor proximity influences surrounding cells. However, we were unable to identify any publicly available NSCLC single-cell datasets that included all of these regions, particularly the tumor-adjacent tissue, which represents the area of greatest interest in our study. As a result, we adopted a compromise approach by analyzing dataset GSE131907 (Data Ref: Ahn & Lee, 2020), which provides paired tumor and distant normal lung tissues, but lacks intermediate tumor-adjacent regions. This limits our ability to directly identify which cell types are altered in response to tumor proximity. Instead, we relied on indirect comparisons and exploratory analyses to infer potential trends. Nonetheless, we find the reviewer's suggestion highly valuable and agree that this question merits deeper investigation. We hope to generate spatially resolved single-cell data in future studies to better elucidate cell-type-specific responses to tumor impact.

To explore which cell types contribute to the observed gene expression changes, we analyzed the expression of *CDKN2A* and *WNT7B* using GSE131907 data. The results showed elevated expression of both genes in tumor tissues, likely driven by increased proportions of basal epithelial cells, ciliated epithelial cells, and stem-like adenocarcinoma cells (Appendix Figure S8). We speculate that the expression of these two genes in the tumor-adjacent areas of some samples was close to the tumor state, which may indicate that stem-like tumor cells spread to the surrounding tissues or that

the surrounding epithelial cells transformed into a tumor-like phenotype. This possibility is supported by our IHC findings, where WNT7B-positive cells at the P5 site exhibited organized epithelial morphology, consistent with transitional epithelial activation. We also noted that normal tissues had an increased abundance of immune cell types—including tumor-associated macrophages, alveolar macrophages, and NK cells—that highly expressed *NOTCH1*, implying a potential immune recruitment process via chemotactic signals. *PDGFRA*, as expected, was specifically expressed in fibroblasts across both tumor and normal tissues, but without significant intergroup differences, indicating likely inter-patient heterogeneity.

Appendix Figure S8. Single-cell RNA analysis of 80576 cells from 10 LUAD patients. (A) UMAP plot of cells colored by major cell types. (B) UMAP plot of cells colored by cell subtypes. (C) Expression levels of *CDKN2A*, *NOTCH1*, *PDGFRA*, and *WNT7B* across different cell types in tumor samples (upper four panels) and normal samples (lower four panels). *FZD10* was excluded from the analysis due to low expression.

To further dissect cell-type-specific contributions on individuals, we applied pseudo-bulk expression profiling after excluding shared cell types within each individual and compared expression differences (Appendix Figure S9). Results revealed that epithelial cells contributed most to *CDKN2A* variation, fibroblasts to *PDGFRA*, and both epithelial and myeloid lineages were major contributors to *WNT7B* expression. Furthermore, clustering based on shallow methylation-associated genes revealed clear tumor–normal segregation primarily within epithelial and myeloid compartments (Appendix Figure S10). These findings suggest that shallow epigenetic transitions may reflect gradual regulatory reprogramming within specific cellular niches, potentially related to EMT, immune modulation, or treatment resistance.

Appendix Figure S9. Impact of cell type exclusion of each patient on target gene \log_2FC . \log_2 fold change of aggregated expression differences for target genes between tumor and normal tissues across patients after excluding specific cell types.

Pseudo-bulk expression profiles were generated by aggregating single-cell RNA-seq data from different cell types within each patient sample.

Importantly, our observations point to the activation or reprogramming of normal epithelial cells at tumor-adjacent sites, potentially driven by DNA methylation alterations, as a key biological process. While the insights derived from GSE131907 remain exploratory due to limitations in sampling and experimental design, we believe that future studies combining spatial transcriptomics, proteomics, and single-cell sequencing—guided by our sampling framework—will help illuminate the cellular mechanisms underlying these gradual methylation dynamics. We have included this part of the results in the discussion of the revised manuscript (in lines 474-498).

Appendix Figure S10. Clustering of gene expression profiles in different cell types. The gene expression patterns of various cell types in tumor and normal samples from different patients are clustered. Each group is a cell type. The upper panel is the situation of all expressed genes, and the lower panel is the expression spectrum clustering of genes with shallow or mixed changing regions.

As just mentioned, in this revision, we aimed to clarify which cell types might contribute to the observed shallow changes by examining WNT7B protein expression in tumor and adjacent tissues using immunohistochemistry (IHC) (in lines 303-324). Consistent with

our earlier findings, the expression pattern of WNT7B protein appeared to follow a shallow trend along the tumor-adjacent tissue direction (Figure 5C-F). Morphological assessment revealed that WNT7B-positive cells in adjacent tissues were predominantly organized, epithelial-like cells, supporting our single-cell transcriptomic findings. While these observations provide preliminary cellular insights, we acknowledge the limitations in our ability to conduct rigorous morphological or pathological classification due to expertise constraints. Future efforts in this direction—particularly those involving collaboration with pathology specialists and the integration of spatial transcriptomics—may offer a more comprehensive understanding of the cellular dynamics underlying these methylation-associated transitions.

Figure 5C-F. Protein expression changes of WNT7B at different locations. (C) Representative IHC staining of WNT7B protein in LUAD from Sample 19. FFPE tumor tissue section was stained using WNT7B antibody. Three directions from the tumor core to adjacent tissues were marked by red lines, and 300μm-squared regions of interest (ROIs) of the tumor core (TC), tumor edge (TE), and adjacent tissue at different distances (P5–Near and P5–Far) on each direction were selected as for optical density (OD) analysis. (D) Three larger ROIs covering the corresponding TE and P5–Near regions were selected for local magnification display of the three boundary regions in Sample 19. HE and H-DAB staining images showed the visual boundary of the tumor

and corresponding DAB signals in TE and P5-Near regions. (E) Violin plots showing the distribution of mean OD per cell for each region (TC, TE, Near, and Far) in three directions of Sample 19. (F) Proportion of OD-positive cells in each region and direction, based on predefined OD thresholds described in Methods.

4. As mentioned in line 102, the authors should provide an example of an HE image in the supplementary figures.

Response:

Thank you for your suggestion. In the revised Figure 1, we have included an HE-stained image showing all seven sampling positions and a representative tumor boundary. This image illustrates the distinct histological differences between tumor and surrounding tissues, and visually highlights the tumor margin referenced in the manuscript (in lines 104-110).

5. A color bar should be included in Figures 2A-2C for clarity.

Response:

Thank you for your suggestion. We have revised the corresponding figures and added color bars to improve clarity and facilitate interpretation.

6. It is recommended to remove the bottom heatmap panel in Figure 5A, as it presents redundant information and may complicate interpretation.

Response:

Thank you for your comment. We have modified the figure as recommended to simplify visualization and avoid redundancy.

Reference:

Ahn M, Lee H Gene Expression Omnibus GSE131907 (2020)
(<https://www.ncbi.nlm.nih.gov/geo/query/acc.cgi?acc=GSE131907>) [DATASET]

Guo W, Zhu L, Yu M, Zhu R, Chen Q, Wang Q (2018) A five-DNA methylation signature act as a novel prognostic biomarker in patients with ovarian serous cystadenocarcinoma. *Clin Epigenetics* 10: 142

Guo W, Zhu L, Zhu R, Chen Q, Wang Q, Chen JQ (2019) A four-DNA methylation biomarker is a superior predictor of survival of patients with cutaneous melanoma. *Elife* 8

Dear Dr. Chen

Thank you for submitting your revised manuscript to EMBO reports and for your patience while it was reviewed. Since Bernd Pulverer is currently traveling, I have taken over the handling of your manuscript.

We have now received the full set of reports (copied below) and as you will see, both referees recommend publication.

Browsing through the manuscript myself, I noticed a few editorial things that we need before we can proceed with the official acceptance of your study.

FORMATTING REQUESTS

- Remove all figures from the manuscript file: main and EV figures need to be uploaded as separate files while Appendix Figures (now Supplementary) are already provided in the Appendix file; the legends of both main and EV figures should stay in the manuscript at the very end while the legends of the Appendix figures need to be in the Appendix file
- Remove the Author Contributions from the manuscript file - you already specified the roles and contributions for each author in our online system
- Funding information in the manuscript file needs to be provided under Acknowledgments and therefore the separate section heading, 'Funding' is not needed
- "Supplementary" should not be used as nomenclature, needs to be updated to the correct one
- Supplementary Tables 2, 3 and 5 are datasets - please update them to Dataset EV1-EV3 in all places (source file names, titles in the system, callouts in the manuscript). Please add the legend in a separate tab of the .xls sheets (see below).
- Supplementary Tables 1, 4 and 6 should be renamed to Table EV1-EV3 in all places (source file names, titles in the system, callouts in the manuscript); these tables and their legends could also be provided in the Appendix file as Appendix Table S1-S3 and the manuscript callouts would need to be updated accordingly
- The legends of the tables need to be removed in the manuscript: for datasets, each legend should be provided as a separate sheet/tab in the Excel file, for EV tables, each legend should be provided in each table above the table while the legends of Appendix tables need to be provided in the Appendix file above each table
- Appendix file needs a title page (title 'Appendix' and not 'Supplementary Figures') that has a table of contents with page numbers: so that the location of each item is shown
- The file "Supplementary Information" does not align with any of our file types. It could be included in the Appendix PDF. Maybe call it "Appendix supplementary methods" or alike.
- Please describe your findings in the Abstract in present tense. This sentence seems an ending "[...] thereby substantiating the regulatory."
- We received bounced email alerts for the following co-author addresses:
pumc_jianchao@student.pumc.edu.cn
kaluk@mail.utoronto.ca
pumc_libowen@student.pumc.edu.cn
These accounts need to be removed from the author list and then added using valid email addresses or you can send us the valid emails and we will update the accounts for these authors
- EMBO press papers are accompanied online by A) a short (1-2 sentences) summary of the findings and their significance, B) 2-3 bullet points highlighting key results and C) a synopsis image in jpeg, TIFF or png that is exactly 550 pixels wide and 200-600 pixels high (the height is variable). The synopsis image should provide a sketch of the major findings, like a graphical abstract. Please note that text needs to be readable at the final size. Please send us this information along with the final manuscript.
- Could the resolution of Appendix (Supplementary) Figure 4 be improved. One needs to zoom in quite a bit in order to be able to read the text, which is possible, but the text starts looking 'fuzzy'
The same is true for Appendix Figure S6. You might want to split this figure into two, to allow showing the panels at larger size.

DATA REQUESTS for figure legends

- Please note that the exact p values are not provided in the legends of figures 2A-C
- Please indicate the statistical test used for data analysis in the legends of figures 4G; 6D, E, F, G, H, I, K; EV5
- Please note that the box plots need to be defined in terms of minima, maxima, centre, bounds of box and whiskers, and percentile in the legends of figures 2A-C; 3A, D; 4F, G; EV1 D, F; EV2 B-G
- Please note that information related to n is missing in the legends of figures 2A-C; 3A, D; 4F, G; 5E, EV1 D, F; EV2 B-G; EV4 C, G
- Please note that the measure of center for the error bars needs to be defined in the legends of figures 5B, EV3

With kind regards,

=====

Referee #1:

The authors have addressed my concerns in their revised manuscript. I do not have additional comments and recommend for publication in EMBOreport.

Referee #2:

The authors have thoroughly addressed all of my previous comments and questions. The revisions have significantly improved the clarity and quality of the manuscript. I am satisfied with the authors' responses and the changes made. I believe that the revised manuscript is suitable for publication in EMBO Reports.

All editorial and formatting issues were resolved by the authors.

Qihan Chen
Macau University
Avenida da Universidade, Taipa, Macau, China
Macau 999078
China

Dear Qihan,

Thank you for approving the final minor text editors. I am very pleased to accept your manuscript for publication in the next available issue of EMBO reports. Thank you for your contribution to our journal.

Kind regards,

Martina
